



# Saharan dust transport event characterization in the Mediterranean atmosphere using 21 years of in-situ observations

Franziska Vogel[1], Davide Putero[2], Paolo Bonasoni[1], Paolo Cristofanelli[1], Marco Zanatta[1], and Angela Marinoni[1]

[1]Institute of Atmospheric Sciences and Climate (ISAC), National Research Council (CNR), Bologna, Italy
[2]Institute of Atmospheric Sciences and Climate (ISAC), National Research Council (CNR), Turin, Italy

**Correspondence:** Franziska Vogel (f.vogel@isac.cnr.it)

**Abstract.** The Mediterranean Basin is regularly affected by atmospheric dust transport from the Saharan desert. These recurring events have strong implications for the Earth's energy budget, cloud formation processes, human health, and solar energy production. Monte Cimone, with 2165 m a.s.l, is an ideal platform to investigate dust outbreaks in Mediterranean Europe. In this study, we present 21 years (2003-2023) of dust transport event identification, derived from continuous measurements of the

aerosol optical size distribution coupled with backward trajectories. Throughout all the years investigated, the fraction of dust transport days remained constant at values between 15 % and 20 % without any detectable trend. This absent trend was also observed in the particulate matter concentration. The annual cycle of dust transport days was characterized by two peaks from May to August and in October and November with values up to 20 %. A similar annual cycle was reflected in the particulate matter concentration with the highest concentrations is summer and the lowest in winter. Grouping consecutive dust transport

days into dust transport events revealed that in the winter months a typical event had a duration of one or two days, whereas in the summer months dust transport events lasted longer (three or more days). The 21 years of measurements presented in this study will set a baseline to assess future dust transport scenarios. Furthermore, they can be used to validate dust forecast models to increase the accuracy of predicting atmospheric dust transport towards the Mediterranean Basin.

## 1 Introduction

Mineral dust has the highest contribution to the global aerosol mass (Kinne et al., 2006), with an annual atmospheric aerosol burden of a few thousand megatons (Kok et al., 2021). It is emitted by wind erosion and resuspension from arid and semi-arid regions across the continents (Knippertz and Todd, 2012). While being suspended in the atmosphere, mineral dust can affect the Earth's energy budget by directly scattering and absorbing incoming radiation (Choobari et al., 2014; Schepanski, 2018). On the other hand, dust particles have the strong ability to form cloud droplets (Karydis et al., 2017) and ice crystals throughout

the entire atmospheric temperature range, leading to a potential full glaciation of a cloud (Froyd et al., 2022; Vogel, 2022). This strongly alters the radiative properties of the clouds and their precipitation capability and therefore influences the Earth's water cycle (Mülmenstädt et al., 2015). Moreover, mineral dust can affect tropospheric chemistry by multiple pathways (van Herpen et al., 2023; Huo et al., 2024; Melssen et al., 2024). Mineral dust deposition enriches the soil and water with nutrients, altering



the oceanic and terrestrial biochemical cycle (Mahowald et al., 2014; Adebiyi et al., 2023); when it is deposited on glaciers
(e.g. in the Alps), it changes their albedo, favoring their melting (Gabbi et al., 2015). Mineral dust has also an impact on human
health, causing respiratory and cardiovascular disorders (Goudie, 2014; Oduber et al., 2019), flight traffic, due to a reduced
visibility (Weinzierl et al., 2012), and solar energy production due to a damping of incoming radiation and dust deposition on
solar power panels (Varga et al., 2024).

The Saharan desert is the largest source region of mineral dust world wide (Miller et al., 2004) and it is still debated whether
its contribution to the atmospheric dust load is increasing (Zuidema et al., 2019) or decreasing (Yuan et al., 2020). Due to their
vicinity, the Mediterranean and Continental Europe are regions frequently impacted by dust outbreaks (Querol et al., 2009; Pey
et al., 2013; Cabello et al., 2016; Gobbi et al., 2019). Hereby, synoptic patterns such as the Mediterranean cyclone, the presence
of an upper-level trough over the Mediterranean basin or anticyclonic conditions associated with convective injection of dust in
north Africa play a crucial role suspending and transporting Saharan dust towards Europe (Brattich et al., 2015; Varga, 2020;
Flaounas et al., 2022). Another important transport pathway over the Saharan desert is the Inter-Tropical Convergence Zone
(ITCZ), a low pressure belt reaching its northern most position over the Sahara in summer, and thus enhancing the dust load in
the atmosphere.

When dust gets enriched in the atmosphere it can strongly increase the particulate matter (PM) concentration. This is true also
in urban areas as shown by measurements carried out in the Po Valley (e.g. Parma, Modena and Cesena) during dust transport
events identified at Monte Cimone, confirming that the contribution of mineral aerosol on the urban PM10 values can be very
critical, favoring threshold exceedance (Bonasoni et al., 2004). Since an enhanced level of the PM concentration can lead to
health issues, guidelines published by the World Health Organization (WHO) give an upper limit of 45 $\mu g\,m^{-3}$ per day for
the concentration of particles with a diameter smaller than 10 $\mu m$. Previous studies, such as Pey et al. (2013); Conte et al.
(2020); Nava et al. (2020); Tositti et al. (2022) investigated the increase in the PM concentration during dust transport events
throughout the Mediterranean region and reported a consistent increase, which is more pronounced in the southern part, closer
to the Saharan desert. Within all the analyzed measurements, there was no consistent seasonal pattern, as some places in Central
Italy had higher PM concentrations in summer, and other places in the winter months (Pey et al., 2013; Petroselli et al., 2024).
One region of interest are the northern Apennines, the first mountain range that air masses from northern Africa cross to reach
central Europe. High-altitude measurement sites are of particular interest, since they are typically not strongly affected by
anthropogenic emissions and can experience both planetary boundary layer and free tropospheric conditions. In particular, the
Monte Cimone (CMN) station, with its location and altitude, has been object of multiple studies to investigate the influence of
Saharan dust transport. Bonasoni et al. (2004) presented the first work on dust, finding a clear correlation between Saharan dust
transport and atmospheric aerosol concentration. Duchi et al. (2016) consolidated this research activity by presenting 10 years
(2002 - 2012) of Saharan dust transport events occurring at CMN to introduce a methodology to identify the dust transport
days by using measurements of the optical particle size distribution and backward trajectory analysis. A commonly applied
approach for dust transport identification is not yet established and methods range from in-situ observations to remote sensing
approaches. Also long-term measurements to validate dust forecast models are still rare.

This work aims at extending the work from Duchi et al. (2016) until 2023, which allows to investigate also trends over two





decades. Furthermore, we provide a detailed analysis of the variability of dust transport days throughout the whole period, and
investigate the variability in the duration of events. We also apply a slightly adjusted approach compared to Pey et al. (2013),
to assess the enhancement in particulate matter due to dust transport. The results presented here can be used for dust forecast
validations and to assess future scenarios of dust transport taking our study as a base line.

## 2 Methodology to analyze and categorize dust transport events

### 2.1 Measurement site and instrumental setup

Monte Cimone (CMN, 2165 m a.s.l.) is the highest peak in the Italian northern Apennines, and is located at 44.19° N, 10.70° E.
The observatory is operational since the early 1990s and is a WMO/GAW (World Meteorological Organization/Global Atmosphere Watch) global station and a national facility of ACTRIS-RI (Aerosol, Clouds and Trace Gases Research Infrastructure;
https://www.actris.eu/) and ICOS-RI (Integrated Carbon Observing System Research Infrastructure; https://www.icos-cp.eu/).
CMN is a remote site, since there are no pollution sources nearby. However, its vicinity to the Po Valley, one of the most polluted urban areas in Europe leads to regular intrusions of pollution (Marenco et al., 2006). In winter CMN is mainly influenced
by air masses from the free troposphere, while in summer it frequently undergoes influence from the planetary boundary layer
(PBL). Due to its altitude, the station can be either inside or outside a cloud. Further details on the measurement site and its
meteorological characteristics can be found in Cristofanelli et al. (2021).

Among other variables, the aerosol optical size distribution is measured with an optical particle counter/sizer (OPC/OPS;
Grimm® model 1.108) since august 2002. Particles in the sampling air enter the instrument and cross a laser light beam. The
90° scattered light of single particles is detected, and depending on the signal intensity, the particles are assigned to one of
the 15 available diameter channels. Hereby, the minimum detectable particle diameter is 0.25 μm and the maximum particle
diameter is 20 μm. The measurements are saved as a particle number concentration per bin with a time resolution of 1 min.
For the analysis in this work, data were averaged over 60 min and later 1 day, with a minimum hourly data coverage of 50 %.
To identify dust transport days (DTDs) we used the coarse particle concentration, i.e., particles with a diameter greater than 1
μm.

The instrument is connected to a heated whole air inlet, which allows for the entry of all aerosol particles and hydrometeors.
The inlet is heated since 2008 (the set temperature is 25 °C) to evaporate any cloud droplets or ice crystals; consequently,
the sampled aerosol, in the presence of clouds, includes both interstitial and residual aerosol. Before this date the temperature
difference between the ambient air the the inside of the laboratory was typically high enough to ensure a relative humidity
below 40 %.



## 2.2 Identification of dust transport events

The method for the identification of dust transport events (DTEs) is based on a pre-selection of potential days using in-situ
measurements of the coarse particle concentration and confirmation by 7 days back-trajectories. While a detailed description
of the method can be found in Duchi et al. (2016), we give a short summary here. The Duchi et al. (2016) approach consists
of four steps. First, the daily average of the coarse particle number concentration, is smoothed with a 21 days moving average,
applied three times to dampen the noise. Second, the third iteration of the moving average is subtracted from the original time
series to obtain the 'high frequency' (HF) component. Third, the days on which the HF component is above the threshold value,
defined as the 95 % confidence interval of all HFs, are flagged as potential DTDs. Fourth, to confirm the pre-selected DTDs, the
source origin of the air parcel must be the Saharan desert. 7-day back-trajectories were calculated with the FLEXTRA model
(Stohl et al., 1995) every 3 h at a starting altitude of 2200 m a.s.l.. For each of the potential DTDs it is checked whether any
of the trajectory points for that day were recognized in a defined grid over North Africa. The grid was a modified version of
the Duchi et al. (2016) study, such that in this work the source contribution was divided into the west, central and east Sahara
as well as the Sahel zone, marked as box 1, box 2, box 3, and box 4, respectively (Fig. 1 c). From all the days fulfilling step
three and four, a final list of DTDs is obtained. To retrieve an unbiased statistics, we only considered months in which the data
coverage was at least 50 %. Furthermore, we retained data only from 2003 on, because OPC measurements from 2002 did not
depict a full year as they started in August.

Throughout the 21 years of measurements presented in this work, 81 % of the data were considered valid. The remaining 19 %
were invalid due to missing measurements or not available back-trajectories. Missing measurements occur due to a malfunction
of the instrument or the instrument being out of service due to routinary maintenance or calibration in the factory. The longest
period of missing data spans over three months.

DTDs are regarded as individual days on which Saharan dust was transported in the atmosphere to CMN. To investigate the
duration of continuous Saharan dust advection, consecutive days were grouped into DTEs. Hereby, consecutive DTDs that
were interrupted by one non-DTD were considered as a unique DTE.

## 2.3 Calculation of PM mass concentration

One of the variables to characterize aerosol load in the atmosphere is the particulate matter (PM) concentration in different
size ranges. Common measures are the PM concentration of particles smaller than 1 μm (PM$_1$), smaller than 10 μm (PM$_{10}$)
and the total PM concentration. For this work we calculated the daily PM concentration of the coarse particles (PMcoarse), in
the same size range as used for the identification of DTDs (particle diameter larger than 1 μm). The daily particle number size
distribution of coarse particles was converted into a mass distribution using a particle density assuming the particle sphericity.
The particle density depends on the particle size and composition. Therefore we applied on our data a particle size dependent
density as presented in Wittmaack (2002).

In our analysis we differentiate the PMcoarse concentration on DTDs and outside of DTDs, the so called 'background'. Note
that the contribution of other events such as pollution or wild fires were not removed from the background conditions. Consid-





ering that these type of particles are predominantly found in the accumulation mode (Lohmann et al., 2016), their contribution to the coarse particle concentration was assumed to be negligible.

## 2.4 PMcoarse enhancement

To assess the enhancement in the PMcoarse concentration compared to the background, we applied the method proposed by Escudero et al. (2007) with the modification reported by the European Commission Staff Working Paper which establishes guidelines for demonstration and subtraction of exceedances attributable to natural sources under the Directive 2008/50/EC on ambient air quality and cleaner air for Europe (https://data.consilium.europa.eu/doc/document/ST-6771-2011-INIT/en/pdf, last access 9 October, 2024). We used the PMcoarse concentration instead of the $PM_{10}$ concentration. The methods consisted

of two steps. In the first step, a 30 days moving average of the background PMcoarse was calculated. In the second step, the enhancement in the PMcoarse concentrations during DTDs was then retrieved from the PMcoarse concentration during individual DTDs and the background PMcoarse. A more quantitative measure on how much dust influences the background PMcoarse concentration is the enhancement factor (EF), calculated as the ratio of the PMcoarse enhancement over the running median of the background.

## 2.5 Trend analysis


To assess the trend in our dataset, we applied the trend detection methodology presented and discussed in detail by Collaud Coen et al. (2020). In short, it combines three different pre-whitening methods to remove autocorrelation and minimize the number of detected false positive trends. In case the dataset has a positive trend, the output is the user-defined alpha value. If no trend was obtained, the output value is 0, -1 or -2, where 0 stands for no trend given from all tests, and -1 and -2 stand for

a false positive test from different tests. This trend analysis was applied on the annual fraction of DTDs and the annual average of the PMcoarse concentration during DTDs.

## 3 Results

### 3.1 Overview of dust transport to CMN

Duchi et al. (2016) presented 10.5 years (from August 2002 to December 2012) of dust transport analysis at Monte Cimone

(CMN), and our analysis extends the time series by a further 11 years until the end of 2023. Figure 1 provides a general overview of (a) the number of dust transport days (DTDs), and (cb) the duration of dust transport events (DTEs) of these 21 years. In this period, 15.8 % of the analyzed days were detected as DTDs following the approach described in Sect. 2.2. These individual DTDs were grouped into DTEs as presented in Sect. 2.2. The majority (42.2 %) of the events lasted one day, whereas the other durations had similar fractions with values of 22.3 %, 15.1 % and 20.4 % for increasing durations. After

the occurrence and duration of DTEs, we investigated the potential source origin of mineral dust within the Saharan desert





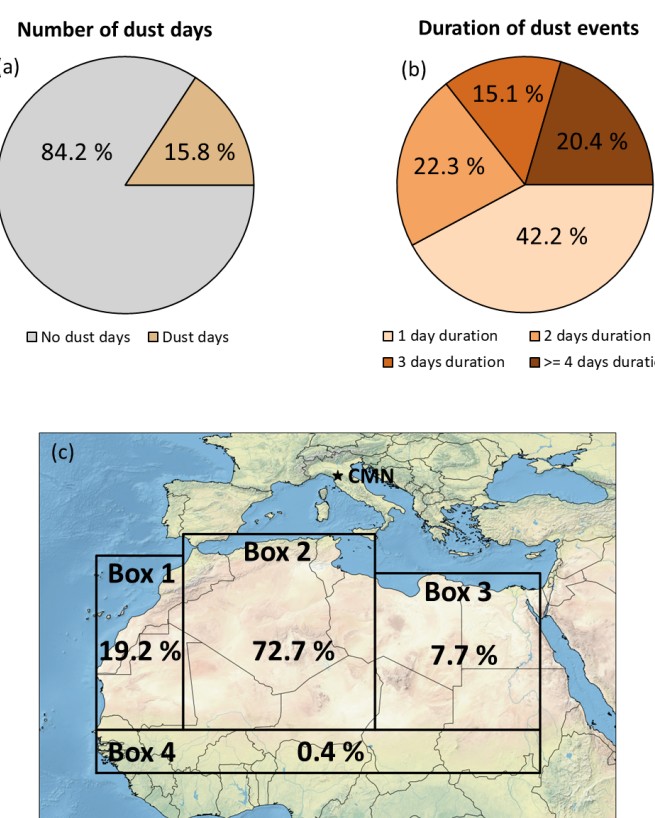

**Figure 1.** Grid box extension for the four boxes used to confirm dust transport days (a). The percentage values give the fraction of time stamps that passed over each box. (b) shows the fraction of dust transport days (brown) and the number of non-dust transport days (grey). (c) shows the duration of dust transport events divided into 1 day (beige), 2 days (orange), 3 days (light brown) and 4 and more days (dark brown). Map made with Natural Earth (naturalearthdata.com).

area (Figure 1 c). The dominant source area was identified to be 'Central Sahara' (box 2), which was crossed by 72 % of all trajectory points across the selected area. While 'Eastern Sahara' (box 3) was associated with 7.7 % of trajectory points, 'Western Sahara' (box 1) was the second most important source region with 19.2 %. Only 0.4 % of the back-trajectory points passed over the southern part of the Sahara, which also includes the Sahel zone (box 4). A similar source contribution from
the different areas of the Sahara is presented in Collaud Coen et al. (2004) and Duchi et al. (2016), where they observed the highest density of trajectories in the northern part of the Sahara, during dust transport to Jungfraujoch and CMN.





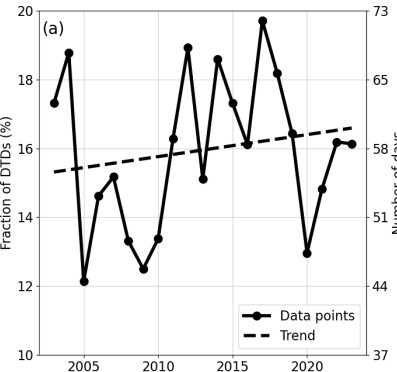 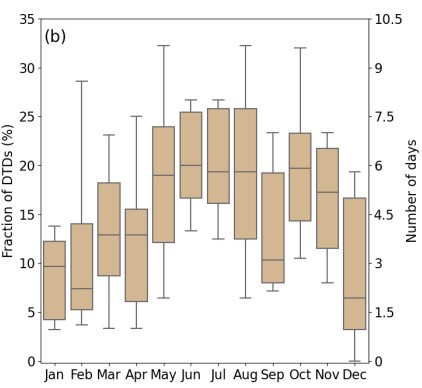

**Figure 2.** Annual fraction of DTDs including the years from 2003 to 2023 (a). The solid line corresponds to the individual data points, and the dashed line shows the trend over all 21 years. The left y-axis gives the fraction of DTDs in relation to the number of valid days per year and the right y-axis is the conversion of the fraction into a number of days per year. (b) monthly fraction of DTDs. The boxes mark the 25th and 75th percentile, while the whiskers are the 10th and 90th percentile. The left y-axis gives the fraction of DTDs in relation to the number of valid days per month and the right y-axis is the conversion of the fraction into a number of days per month.

### 3.2 Frequency of dust transport at Monte Cimone

The annual fraction of DTDs was calculated as the ratio of the annual number of DTDs over the total number of valid days

per year (Fig. 2 a). On average, the annual fraction of DTDs was 15.8 %, meaning that CMN was affected by Saharan dust transport on about 58 days per year. The fraction was fluctuating between 12 % and 20 % with multi-annual periods of lower or higher fractions. Within the 21 years of dust identification, there was no significant temporal trend (slope of 0.063) in the fraction of DTDs obtained from the trend analysis described in section 2.5. The quantification of the trend might be affected by the high fractions at the beginning and the low fractions at the end of the time series as well as the lower and higher fraction

from 2005 to 2010 and 2012 to 2017, respectively. Saharan dust is transported by large-scale synoptic patterns, such as the Mediterranean Cyclone, which change in position and intensity throughout the year, and thus influence the seasonal variation of DTDs, but not the interannual variability (Varga, 2020; Flaounas et al., 2022).

The annual cycle was investigated merging all the years for each month (Fig. 2 b), and revealed a clear cycle. A broad maximum in the median fraction of DTDs up to 20 % was observed for May, June, July and August, which was followed by a secondary

maximum in October and November, with similar high median fractions. The winter months (December, January and February) showed a minimum fraction with median values from 6 % to 10 %. This means that in the summer months CMN experienced Saharan dust transport on about 6 days per month, while in winter this number was reduced to 2.5 days. The interannual variability, depicted by the whiskers, does not follow an annual cycle and is rather driven by one or two years that showed a comparably very high number of DTDs in a specific month. This variability reached values from 0 % in December, meaning

that no DTDs were detected in at least one year, up to 32 % in May, August and October. The annual cycle in DTDs is consistent with Pey et al. (2013), who gave a monthly probability of DTDs for central Italy similar to our measurements with a high peak





in May, June and August and a secondary peak in October and November. Also, a modeling study by Israelevich et al. (2002) suggests an annual cycle with a high aerosol index in summer over the central Mediterranean region. However, Petroselli et al. (2024) observed in a low mountain site in central Italy a rather inverse trend compared to our measurements with a minimum in July and August and a maximum in the winter months. This could be due to the fact the site in their study is at a lower altitude (1100 m a.s.l.). By that it experiences a different impact from the boundary layer dynamics and might not be in the free troposphere as often as CMN. Moreover, they applied the Duchi et al. (2016) approach on hourly data, which can lead to different results.

Monthly changes in the fraction of DTDs can potentially reflect the location of the Mediterranean cyclone, which in the summer months occurs preferably over the north-western part of Africa, the Atlas mountain, and thus enhances dust transport towards Italy (Varga, 2020; Flaounas et al., 2022). This could also explain, that the major source region of Saharan dust is the Central Sahara. Another large scale synoptic pattern, that could contribute to the enhanced fraction of DTDs from May to August is the Inter-Tropical Convergence Zone (ITCZ). Its position in the summer months is at around 20°N and by that can enhance the northward transport of dust loaded air (Ginoux et al., 2001; Sunnu et al., 2008). In winter, however, it is positioned around 5°N, which prohibits dust transport. The reason for the second peak in October and November is yet unclear, however, a possible explanation could be found in Medicanes, mainly occurring in these months, which potentially enhance the transport of dust towards southern Europe.

### 3.3 PMcoarse concentration during and outside dust transport days

### 3.3.1 Recurring interannual cycle

The PM concentration is a regulated air quality variable that describes the atmospheric aerosol burden in terms of mass, which helps to quantify the level of pollution of ambient air. Previous studies such as Querol et al. (2009); Pey et al. (2013); Petroselli et al. (2024) made use of the PM10 concentration to assess the contribution of Saharan dust to the background PM10 concentration. As presented in Section 2.3, we calculated the PM concentration of the coarse particles only (PMcoarse) during and outside DTDs. The background PMcoarse concentration showed median values between 0.3 µg m$^{-3}$ and 3 µg m$^{-3}$ (Fig 3 (a), gray line). Within the whole observation period, the median PMcoarse concentration on DTDs (Fig 3 (a), brown line) was about one order of magnitude higher than the background conditions. Also the 25$^{th}$ percentile of the PMcoarse concentration on DTDs was in almost all years higher than the 75$^{th}$ percentile of the background concentration, meaning that the increase of PMcoarse during DTDs was relevant. In the study by Millán-Martínez et al. (2021) they also observed enhanced PM10 concentrations during dust transport episodes, but the difference to the background was reduced to a factor of 1.5. Both, the PMcoarse background and during DTDs, showed a wave-like profile with a wavelength of about 12 years. Minima were observed in 2006 and 2019 - 2020, while a broad maximum from 2011 to 2013 was reached. This wave-like pattern was reported the first time and a potential connection to atmospheric circulations like ENSO (El-Nino Southern Oscillation) or NAO (North Atlantic Oscillation) could not be confirmed. A trend analysis performed on the PMcoarse concentration during DTDs revealed no trend for this data set, which can be connected to the wave-like pattern. The annual enhancement, calculated as



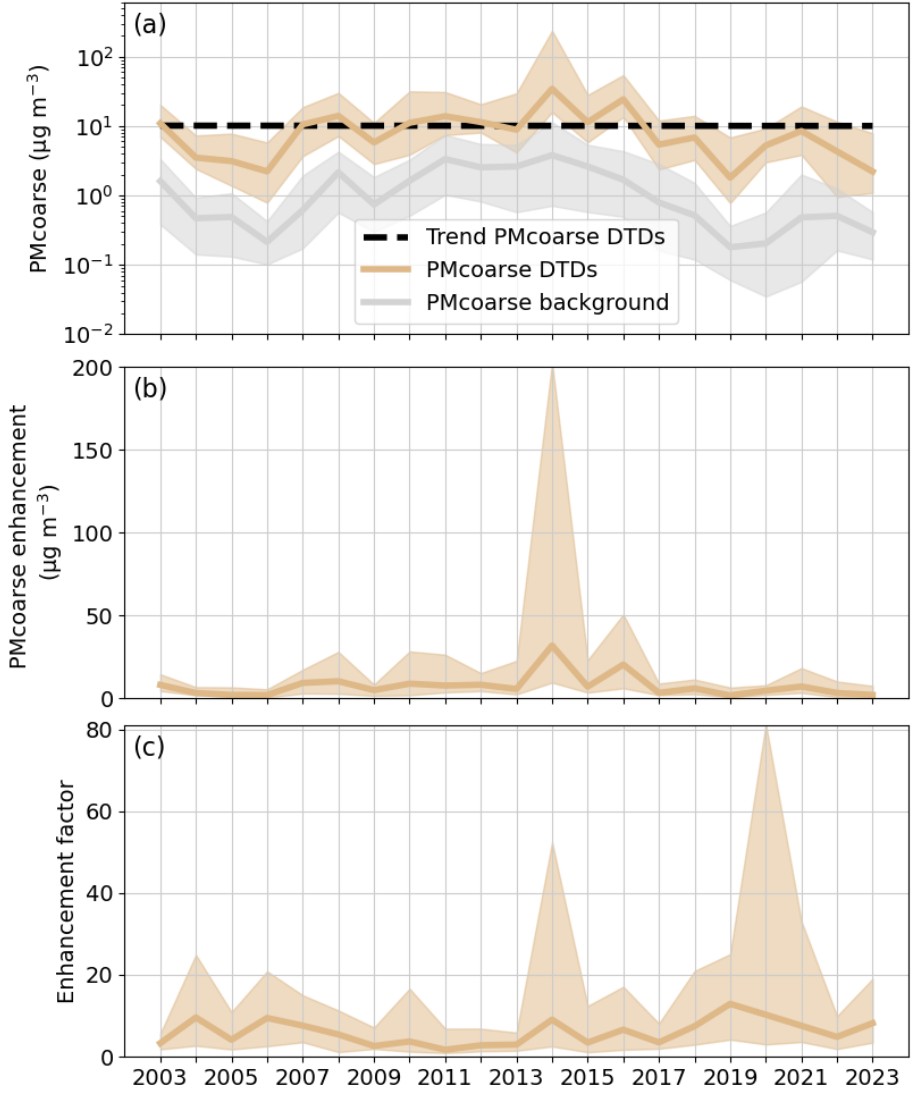

**Figure 3.** (a) Annual PMcoarse concentration during (brown) and outside (grey) DTDs. The dashed line shows the trend in the PMcoarse concentration during DTDs. (b) Enhancement in the PMcoarse concentration during DTDs. (c) Enhancement factor (EF) of the PMcoarse concentration during DTDs. In all panels, the solid line shows the median values; the shaded area around is the 25th and 75th percentile.

described in section 2.4, was very variable and showed median values between 2 $\mu g\,m^{-3}$ and 33 $\mu g\,m^{-3}$ (Fig. 3 b). Some years showed a higher variability, where the 75$^{th}$ percentile reached values up to 50 $\mu g\,m^{-3}$, or in the extreme case of 2014 up to 200 $\mu g\,m^{-3}$. This can be caused by one or two extreme DTEs with very high PMcoarse concentrations, as it can also be seen in Figure 3(a), where especially in 2014 it reached the highest median concentrations. The annual enhancement factor (EF) varied between median values of 3 and 12, with an overall median value of 7 (Fig. 3 c). Higher EFs, such as in 2020, can




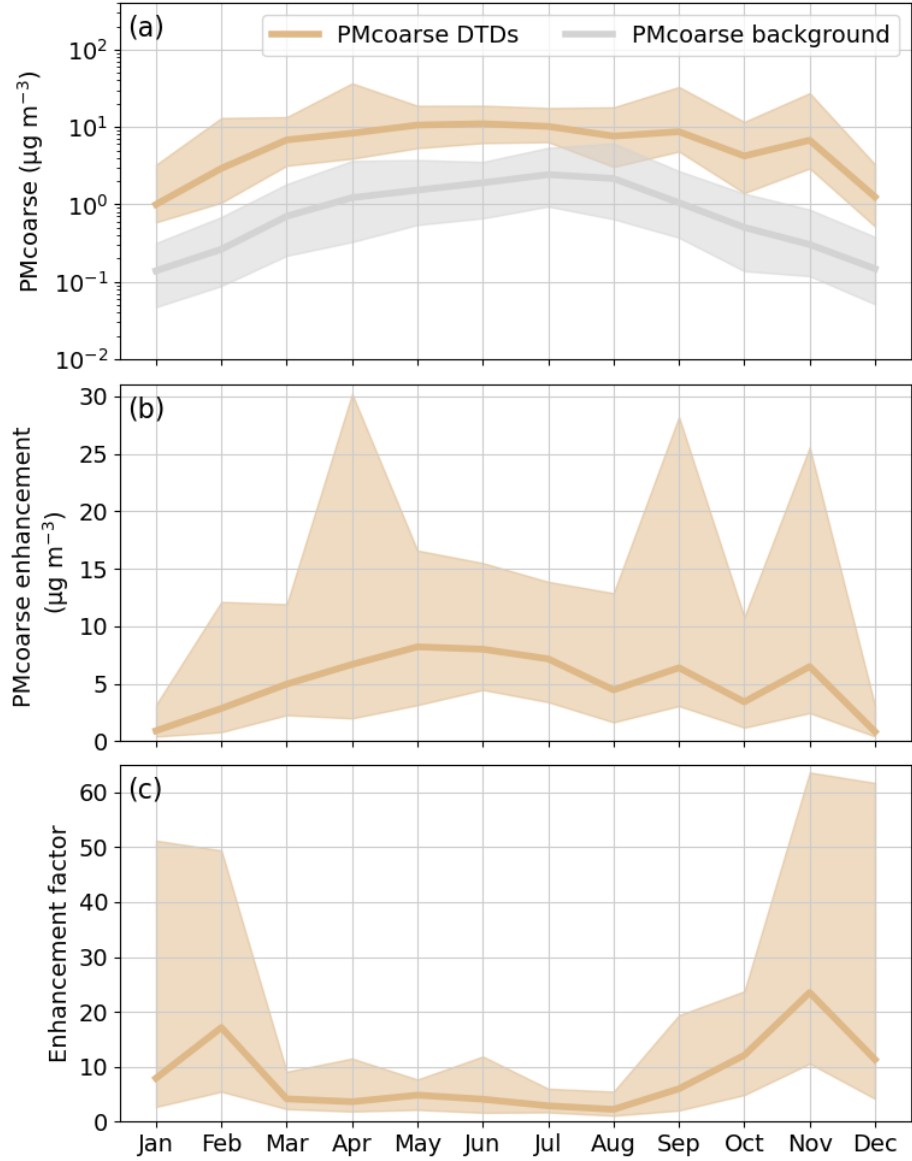

**Figure 4.** (a) Monthly PMcoarse concentration during (brown) and outside (grey) of DTDs. (b) Enhancement in the PMcoarse concentration during DTDs. (c) Enhancement factor (EF) of the PMcoarse concentration during DTDs. In all panels, the solid line shows the median values; the shaded area around is the 25th and 75th percentile.

originate from DTEs during which the PMcoarse background was very low and the PMcoarse concentration on DTDs was high.





### 3.3.2 Seasonal cycle

The PMcoarse background concentration (Fig. 4 (a), gray line) showed a clear annual cycle with minimum median values of about 0.2 µg m$^{-3}$ in winter and a maximum median value of about 3 µg m$^{-3}$ in summer. The PMcoarse concentration during DTDs (Fig. 4 (a), brown line) followed this annual cycle of the background and was almost all of the time one order of magnitude higher, with minimum median concentrations in winter of about 1 µg m$^{-3}$ and maximum median concentrations in summer of about 10 µg m$^{-3}$. One exception is November, where the concentration during DTDs was strongly driven by one extreme event in 2014. Escudero et al. (2007) and Querol et al. (2009) observed in their studies a similar annual variability in Spain and the Mediterranean basin with higher concentrations in summer compared to winter. This result supports the idea that in winter, when CMN is more often in the free troposphere, background concentrations are lower while in summer, when CMN is most often affected by PBL air masses, background concentrations are higher. Other than the effect of the PBL, also wet removal by precipitation can promote the observed annual cycle. The lack of rainfall in summer hinders wet removal (Mifka et al., 2022) and by that aerosol particles in the atmosphere and specifically the PBL are enriched, increasing the background values as well as the dust concentration (Wang et al., 2021). The enhancement in the PMcoarse concentration followed the same annual cycle as the PMcoarse concentration (Fig. 4 b), which is a reasonable behavior as the dust inputs act as a flux that is superimposing to the background. It was the lowest in January and December, with median values of 1 µg m$^{-3}$. These two months also showed the lowest variability, since the spread between the 25$^{th}$ and 75$^{th}$ percentile is 3.5 µg m$^{-3}$ at maximum. In spring and early summer, the PMcoarse enhancement was much higher with maximum median values of up to 8 µg m$^{-3}$. Also the variability within one month was increased compared to winter and the spread between the 25$^{th}$ and 75$^{th}$ percentile reached up to 25 µg m$^{-3}$. The EF (Fig. 4 c) showed an opposing profile compared to the PMcoarse concentration and enhancement, with a minimum median value of about 2 and a low variability in spring and summer, and maximum median values between 15 and 25 with a high variability from September to February. Due to the cleaner atmosphere in the winter free troposphere, the EF during DTEs is higher compared to the enhancement dust induces in summer, in addition to the aerosol population in the PBL.

### 3.4 Interannual variability and annual cycle in the duration of dust transport events

As described in section 2.2, the identified DTDs may be grouped into DTEs of various duration. In this study, we chose 4 different durations, namely 1 day, 2 days, 3 days and $\geq$ 4 days. The fraction shown in figure 5 describes the number of DTEs for each duration group over the total number of DTEs in the respective year or month.

The interannual variability of these four duration groups showed fluctuations throughout the 21 years, but no distinct pattern (Fig. 5 a). In most of the years, the highest fraction was observed for 1 day events, with values between 30 % and 55 %. The second highest fraction in the majority of the years was 2 day events with values between 15 % and almost 20 %. Exceptions are the years from 2012 to 2018, in which the 2 day events represented the smallest fraction. The last two groups of 3 days and $\geq$ 4 days made up a fraction between 5 % and 35 % each. However, there was no tendency towards longer of shorter DTEs



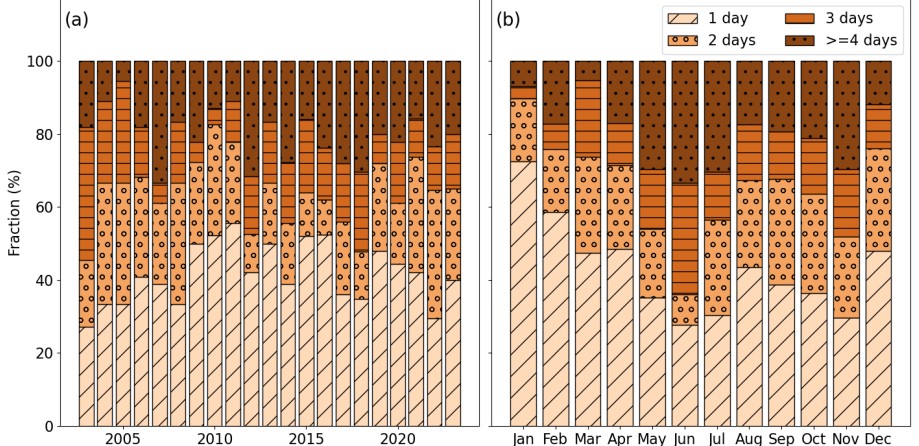

**Figure 5.** Fractional duration of DTEs on an annual scale (a) and a monthly scale (b). The duration is divided into 1 day (beige with diagonal stripes), 2 day (orange with circles), 3 day (light brown with horizontal stripes) and ≥ 4 days (dark brown with dots) events. For each year and month the fraction is calculated as the ratio of the number of DTEs in the various duration groups over the total annual or monthly number of DTEs.

over the years.

When looking at the annual cycle, clear changes in the duration of DTEs were visible (Fig. 5 b). The fraction of 1 day DTEs decreased from 75 % in January to 30 % in June, slightly increased afterwards and showed a second, equally low, minimum in November. The fraction of 2 day events was constant throughout the months with values around 15 %, except June where the fraction decreased to 5 %. As for the 2 day events, also the 3 day events showed a constant fraction around 10 %. Exceptions

here were January and February, with lower fractions of 2 % and 5 % respectively and June with a higher fraction of 25 %. The class of the longest DTEs of ≥ 4 days was the highest between May and November with 20 % to 30 % and the lowest in the winter months and March, when it decreased to 5 %.

The study of Duchi et al. (2016) already suggested a similar seasonal cycle for DTEs, with the majority of the winter events being of 1 day duration and the summer events being more often of a duration of multiple days. Also Petroselli et al. (2024)

reported slightly more DTEs of 1 day duration in winter compared to summer, but did not observe in increase in the duration of DTEs in summer.

Differences in the fractions of the different duration groups, between January and June, could originate from the increased dust mobilization in summer(Vandenbussche et al., 2020; Mousavi et al., 2023) and PBL height over the desert. In summer, the PBL can reach up to 6 km (Knippertz and Todd, 2012), so dust resides in the atmosphere above the Sahara at already high altitudes.

As soon as atmospheric circulation is favorable, dust transport towards the Mediterranean will occur and depending on the stability of the circulation it can last multiple days. In winter, when the PBL height over the Sahara is lower, dust intrusions to altitudes of long-range transport are more prohibited. Moreover, during winter CMN is more often in the free troposphere





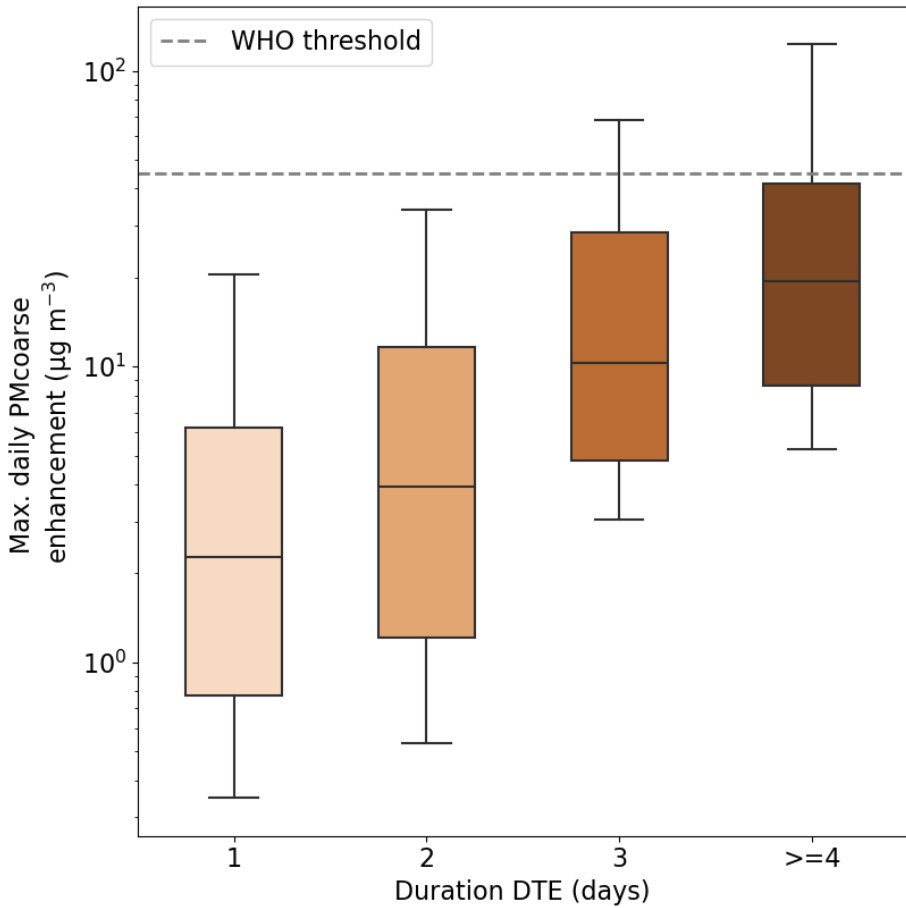

**Figure 6.** Maximum daily PMcoarse enhancement for the four different lengths of DTE.

where air masses can be diluted much faster.

## 3.5 Higher PMcoarse enhancement during longer events

To assess the intensity of DTEs as function of their duration and peak PMcoarse enhancement, we compared the maximum daily PMcoarse enhancement of the different DTE durations (Fig. 6). The median of the maximum daily PMcoarse enhancement increased steadily with the duration of DTEs from about 1.5 µg m$^{-3}$ for 1 day events to about 15 µg m$^{-3}$ for DTEs that lasted at least 4 days. This means that the longer the DTE, the more likely it is to reach higher enhancements in the PMcoarse concentrations. The same pattern was observed when dividing the data into the different seasons (Figure 7). One reason for the observed behavior could be that for longer DTEs a more extensive dust plume reached CMN, which might be connected to a dust storm over the Saharan desert induced by strong winds. Short DTEs might originate from dust transport of the generally



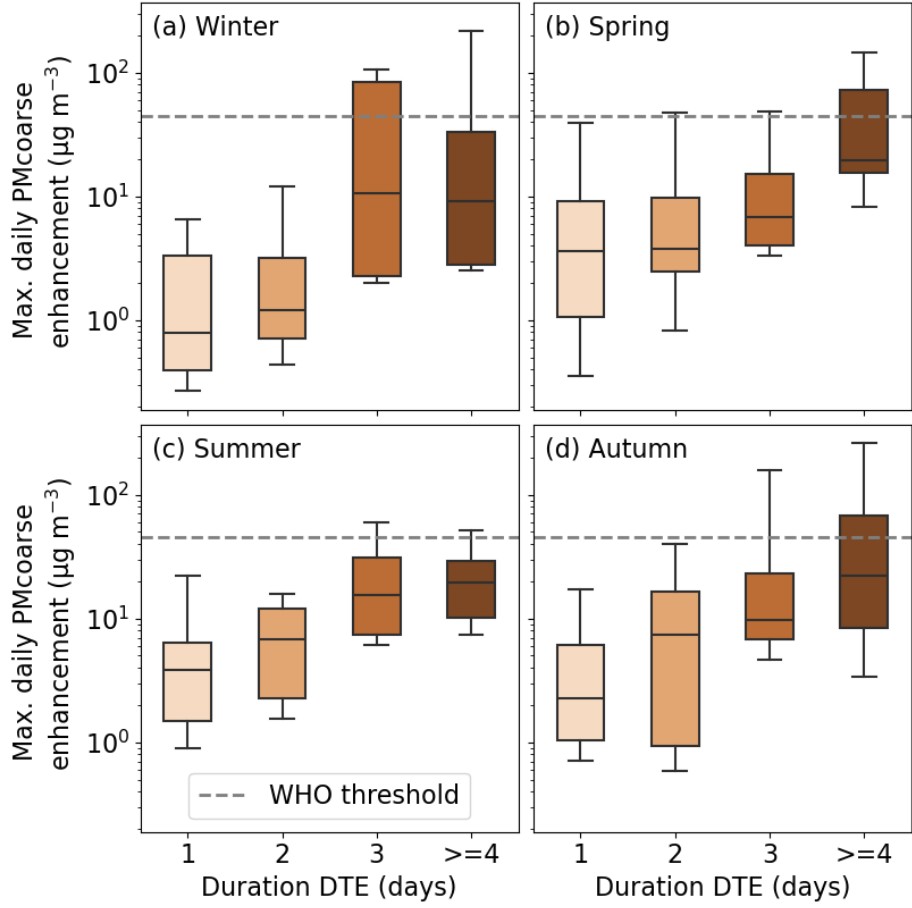

**Figure 7.** Maximum daily PMcoarse enhancement for the different durations of dust transport events. The data are divided into the four seasons: (a) winter, (b) spring, (c) summer and (d) autumn.

dust loaded air over the Sahara without a prior dust storm. Another important point to mark here is that during longer DTEs it seems more likely that the PMcoarse enhancement exceeds the threshold value of 45 µg m$^{-3}$ given by the WHO. Even though

CMN is at 2165 m a.s.l., this accounts in our case for almost 10 % of all detected DTDs. The numbers give only a lower limit as we consider the PMcoarse and not the PM10 concentration.

## 4   Conclusions

In this paper we present 21 years, from 2003 to 2023, of continuous measurements of the optical particle size distribution at the mountain station of Monte Cimone (CMN), the highest peak of the Italian northern Apennines, from which we identified

Saharan dust transport events (DTEs). The data showed a similar high fraction of dust transport days (DTDs) of 15.8 % throughout all the years, with no detectable increasing or decreasing trend. We suggest that the missing long-term trend could



possibly originate from the fact that the Saharan desert is a persistent source for a time exceeding the duration of our study, thus changes in the emission intensity may occur on longer time scales. The annual cycle of DTDs was characterized by one broad maximum in summer and a secondary maximum in October and November, with equal high fractions of DTDs of about 20 %. For the annual variation in DTDs, the position and strength of the Mediterranean cyclone plays a crucial role together with the position of the ITCZ, where a Mediterranean cyclone placed over northwest Africa and the more northward position of the ITCZ in summer time promote dust transport.

We show that dust transport consistently increases the PMcoarse concentration by one order of magnitude compared to the background. Our findings on the annual variation in the PMcoarse concentration report a higher enhancement factor in winter compared to summer, as in winter the background concentration is very low and a DTE represents a major disturbance in that season.

The variation in the duration of DTEs, with longer lasting events in summer than in winter, could have three main reasons: (i) due to a higher planetary boundary layer height in summer over the Sahara, also the dust injection is higher and by that dust is more easily transported towards the Mediterranean when atmospheric conditions are favorable (Merdji et al., 2023); (ii) an increased dust mobilization in the summer months (Vandenbussche et al., 2020; Mousavi et al., 2023); and (iii) different transport processes in summer compared to winter. While in winter, CMN is more frequently in the free troposphere, and exposed to faster air mass transport, in summer CMN is more often affected by PBL air masses, where the mineral dust transported from Northern Africa can reside longer. Moreover, wet removal of aerosol particles is reduced during the dry summer months compared to winter, leading to a longer residence time in the atmosphere.

With our measurements, we provide valuable information on Saharan dust transport over Italy, which might directly impact the energy sector and its solar energy production. As, on average, 58 days per year are affected by Saharan dust transport, we can emphasize that strong effects are expected even far from the source region, with an enhancement in the PMcoarse concentration between 1 µg m$^{-3}$ and 8 µg m$^{-3}$, with an occasional exceedance of the WHO threshold of 45 µg m$^{-3}$.

This study set a milestone in DTE identification, providing a robust and long dataset based on the mass concentration, which will help in validating the rising number of dust forecast products provided by intergovernmental entities such as Copernicus. Future studies can be directed towards (i) investigating in how far wet removal of dust on its transport pathway influences the presence and amount of dust reaching CMN or other locations (Mifka et al., 2022), (ii) combining the here applied method with other methods using e.g. the optical properties of dust (Collaud Coen et al., 2004) and (iii) investigating the various lengths of DTEs by using the optical depth of reanalysis data from satellites over both the Sahara and CMN. This will help to improve our understanding of dust transported in the atmosphere under changing Mediterranean conditions.

*Data availability.* The FLEXTRA trajectories used for the detection of dust transport events can be accessed under https://nadir-trajectories. nilu.no/trajectories/ by selecting 'modelldata', the respective year and then the station 'mtcimone' (Last access: Nov 7, 2024). The OPC data will be made available with a DOI upon final publication.



*Author contributions.* FV analyzed and interpreted the data and wrote the manuscript. DP supported the data analysis. DP, PB, PC, MZ and
320 AM contributed to the interpretation of the data set and the discussion of the results. All authors reviewed the manuscript.

*Competing interests.* The authors declare that they have no conflict of interest.

*Acknowledgements.* We would like to thank Maurizio Busetto, Francescopiero Calzolari and Fabrizio Roccato for their valuable work for
the instrumental maintenance, data acquisition and data flow. The authors gratefully acknowledge the Italian Air Force (CAMM) for access
and logistic support at Monte Cimone.

*Financial support.* This work was supported by the EU-funded EUSAAR PF6 (European Supersites for Atmospheric Aerosol Research),
ACTRIS and ACTRIS-2 H2020 (Aerosols, Clouds and Trace gases Research InfraStructure Network), and by the Italian Ministry for Uni-
versity and Research (MUR) through PON PER_ACTRIS_IT and the ITINERIS projects. The Italian component of ACTRIS RI was also
funded on national level by Fondo Ordinario per gli Enti di ricerca (FOE) for ESFRI activities. FV was funded by 'Progetto nazionale Raf-
forzamento del Capitale Umano CIR01_00015 – PER_ACTRIS_IT Potenziamento della componente italiana della Infrastruttura di Ricerca
Aerosol, Clouds and Trace Gases – Rafforzamento del Capitale Umano'.



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
