# Peer review of "Saharan dust transport event characterization in the Mediterranean atmosphere using 21 years of in-situ observations"

_EGUsphere, 2025_

## Author Comment (AC1)

**General comment**

The manuscript by Vogel et al. (2025) present the analysis of 21 years of optical particle counter (OPC) in-situ observations at the remote Monte Cimone site as an extension of the work by Duchi et al. (2016). The authors analyse the OPC data to provide an estimate of the trend for the annual fractions of dust transport days (DTDs) and PMcoarse, describe the PMcoarse variability compared to background values (enhancement factor) and investigate the correlation between dust events and PMcoarse concentrations.

From a general point of view, although the paper is well written, the analysis could benefit from further refinement and inclusion of additional data sources (i.e. satellite data or reanalysis) and a more accurate and clear presentation of the results (e.g. discussion of measurement uncertainties), which I see as the added value of this work compared to Duchi et al. (2016). I would recommend considering publication once the comments have been adequately addressed.

We thank Reviewer 1 for the detailed and useful comments. Based on that we made the following main changes in the manuscript:

- Added a section of detailed description of the backward trajectories
- Added a section discussing the uncertainties
- Added a paragraph comparing our work to Duchi et al. (2016)
- Added two tables to summarize the results
- Added supplementary material

The structure of our answer to the comments is: black – Comment of the reviewer, blue – Author's response, red – changes made in the manuscript

**Specific comments**

- This work appears to be an extension of Duchi et al. (2016). How do the results of your analysis compare quantitatively with those presented in that study? This comparison is missing in the paper. Which is the added value compared to that study? Please discuss.

  We agree with the reviewer that a part dedicated to the comparison to Duchi et al. (2016) is missing. While both studies give a general overview of the fraction of dust transport days and its seasonal cycle, they differ in the subsequent analysis of these events. Next to the identification of dust events, Duchi et al. (2016) analyze the source origin of the dust and the relation to the coarse particle number concentration. The use of number concentration, however, limits the comparability with other studies. Hence, in our work we converted the size distribution into PMcoarse mass concentration, to better compare our data to previous studies based on in-situ measurements. Considering the comments of both reviewers, the updated version of our manuscript includes a detailed evaluation of uncertainties, which helps quantifying the reliability of the current approach, which was not detailed in Duchi. Overall, the present manuscript provides a more complete phenomenological context and uncertainty evaluation of dust events in

Europe, representing an evolution and not only an extension of Duchi. To point this out we added a paragraph in L.192, which reads as:

Comparison to the study of Duchi et al. (2016)
Duchi et al. (2016) analyzed the dust transport at Mt. Cimone between 2002 and 2012. The dataset in this paper extends this analysis until 2023. Both studies observed an overall fraction of DTDs of 15.7 % or 15.8 %, indicating that the annual fraction of DTDs did not change significantly. Also, the seasonal cycle of DTDs was consistent in both studies, with a broad maximum in spring/summer, a second maximum in October/November, and a minimum in winter. When looking at the duration of DTEs, the highest fraction was always the 1 day duration events with 44 % for Duchi et al. (2016) and 42.2 % in this study. For Duchi et al. (2016) the second highest fraction with 28 % were the 2 day events and further they only report that 8 % of the DTEs lasted more than 5 days. In this study, the fraction of the 2 day events was reduced to 22.3 %. The further duration classification differed slightly, as we categorized differently the DTEs based on their duration. After the discussion of the occurrence of DTDs and the seasonal cycle, Duchi et al. (2016) focused their work on the changes in the coarse particle concentration during DTDs and the source origin from the various parts of the Saharan desert. In our work we discuss the interannual variability and the seasonal cycle of the PMcoarse concentration instead of the coarse particle concentration, so that our results can be more comparable to other studies. Furthermore, we give an estimate of the uncertainty related to this analysis.

- I would see the added value extending this analysis to include spatially-resolved data such as satellite data and/or reanalysis, especially considering that Monte Cimone is located at 2165 m a.s.l., where aerosol optical depth can provide additional insights into dust episodes. OPC in-situ observations alone may not be sufficient to provide a robust estimate of trends in DTD/DTE and to set a "milestone in DTE identification".

The main objective of this manuscript is to extend the analysis of Duchi et al (2016) and to provide further variables from the OPC in-situ observations at Monte Cimone. As of our knowledge, this is the longest record of OPC data from which dust transport days can be assessed. Current efforts aim to integrate our measurements with FLEXPART (https://flexpart-request.nilu.no/data-access) and CAMS reanalysis products. This work will allow characterizing sources, transport duration and vertical distribution of dust in the region. The integration activity is in progress and will be subject of future publications. Considering the wide spread of OPCs on monitoring networks, our in-situ approach and integration with modelling products may be implemented to other stations, extending the spatial resolution of dust event characterization to the full Mediterranean area. In view of the complexity of the analysis presented here and ongoing work on FLEXPART and CAMS products, we did not include additional analysis on remote sensing data.

What happens at your analysis if you neglect the data before the 2008 (i.e. after the inlet in the line has been heated)? I rather see a decreasing trend from 2008 to 2023 (Fig. 9) in PMcoarse and a more increasing trend in DTDs (Fig. 2). Please detail and consider adding supporting information.

The point made by the reviewer is valid, since excluding or focusing on a specific temporal interval may allow to study shorter term tendencies on natural aerosol caused by, for example, large scale meteorology (https://doi.org/10.1029/2023JD039592). An increasing tendency in both DTDs and PMcoarse concentration may be identified till 2014, while a decreasing trend could be identified afterwards. However, no correlation was found with large scale events such as the North Atlantic oscillation and the Western Mediterranean oscillation. Therefore, short- and medium-term tendencies were not calculated.

We do realize that a change in sampling conditions may introduce a bias in the trend analysis. Between 2002 and 2007 the measurements were performed behind a downward facing inlet, without active heating and sampling at a reduced flow of less than 20 LPM. In 2008 a more modern inlet sampling 150 LPM was installed at CMN and equipped with heating system (25°C). Due to the higher sampling flow, increasing the temperature at the top of the inlet line ensured the evaporation of droplets, which may be lost due to sedimentation or impaction in the sampling line. In the inlet configuration between 2002 and 2007, the low sampling flow and passive heating from the room air maintained the inlet line at higher temperature compared to ambient conditions, ensuring evaporation of. The measurements in the inlet line performed in 2007 indicate that passive heating from room air was enough to increase the sampling line temperature, compared to ambient conditions by approximately 15 K, leading to a decrease of relative humidity between 0% and 80%. The increase in temperature allows to drastically reduce the relative humidity in the sampling line below 45% in clear-sky and cloud conditions. Overall, only 13% of measurements performed in 2007 showed a RH between 40-45%. These conditions answer to ACTRIS-RI recommendations, and do not introduce bias in the sampling and measurement of aerosol particles. Hence, we are confident that the change of the inlet in 2007 did not cause a bias in the coarse particle measurements.

We do agree that this was not properly explained in the manuscript. Hence, we changed the text in L.82ff to:

"The instrument is connected to a whole air inlet, which underwent important modifications during the 21 years of measurements presented in the manuscript. Among the changes was the implementation of a heating system at the top of the sampling line in 2008, to better control the humidity in an increased sampling flow (150 l min$^{-1}$). Due to the smaller sampling flow in 2002-2007 (below 20 l min$^{-1}$), the passive heating of the room maintained a warmer temperature in the sampling line ensuring RH values below 40%, as suggested by ACTRIS-RI sampling guidelines. More details are provided in the supplementary material S2"

A text in the supplementary material, including Figure 1 and 2 was introduced to detail the inlet variation.

The conditions of sampling are extremely important since they control particle transmission efficiency. A full set of recommendations is provided by ACTRIS-RI in the following document: https://www.actris-ecac.eu/files/ACTRIS_standard_procedures_for_aerosol_in-situ_measurements.pdf (last accessed 02/07/2025). At CMN, from 2002 to 2007, aerosol sampling was conducted using a downward-facing inlet without active heating, operating at a low flow rate of <20 LPM. In 2008, a new and larger inlet was installed to accommodate more aerosol measurements, featuring a heated design (set at 25 °C) and a higher flow rate of 150 LPM. During the earlier sampling period (pre-2008), although active heating was not used, the low sampling flow allowed for passive heating by room air. This passive heating effectively raised the temperature of the inlet line above ambient conditions, keeping the relative humidity (RH) sufficiently low to enable droplet evaporation Measurements conducted in 2007 at the end of the inlet line indicate that the sampling line was approximately 15 K warmer than the ambient, leading to a significant RH reduction, up to - 80% (Figure S1). Overall, only 13% of data collected in 2007 had RH between 40–45%, indicating that the system operated under conditions suitable for aerosol sampling, in both clear-sky and cloud conditions (Figure S2), following ACTRIS-RI recommendations.

[Figure]

Figure 1 Histogram representing the change in temperature (T) and relative humidity (RH) in the sampling line compared to ambient conditions. Data represents the sampling line and ambient conditions at the Monte Cimone observatory in 2007.

[Figure]

Figure 2 Relative humidity measured in the sampling line at the Monte Cimone observatory in 2007 during cloud and clear-sky conditions.

- You present a dataset without discussing the uncertainties in the measurements/plots. This aspect must be better clarified and taken into account in the statistical analysis/plots avoid limiting the analysis of dataset variability to percentile-based metrics only. How significant is your trend estimation considering the uncertainties you have in the measurements?

  A similar point was raised by reviewer 2, so we included in the method section a detailed paragraph on the instrumental uncertainties. Moreover, their influence on the data is discussed in multiple parts throughout the manuscript. In the following we list the changes made in the manuscript:

  - UNCERTAINTY DEFINITION: we added in the methods a section to describe the uncertainties of the individual terms in the calculation of the PMcoarse concentration. The new section in the methods reads as follows:

  2.7 Uncertainties

  The calculation of the PMcoarse concentration is subject to uncertainties. Given Equation 1, individual uncertainties of the particle diameter ($d_i$), the particle number concentration ($C_{n,i}$) and the particle density ($\rho_i$), are propagated into a final uncertainty of the PMcoarse concentration.

  For $C_{n,i}$, the manufacturer provides for the same OPC model an uncertainty between 3% and 5%. In the literature, the characterization of the uncertainty is limited to one study by Burkart et al. (2010) who observed a 9 % higher total number concentration measured by the same OPC compared to a differential mobility analyzer. However, they did not convert the electrical mobility diameter to an optical equivalent diameter, which can

lead to an increased uncertainty. We therefore apply in our calculation of the error propagation the uncertainty of 5 %.

An uncertainty for $d_i$ is not provided by the manufacturer of the OPC; however, it should be accounted due to biases in the correct sizing introduced by non-spheric particles. Putaud et al. (2004) suggest in their study at Monte Cimone a particle sizing uncertainty of 10 % outside of DTDs and of 20 % during DTDs. The higher uncertainty during DTDs arises from the high degree of non-sphericity of dust particles.

The uncertainty for ($\rho_i$) is not given in the study by Wittmaack (2002), which we used to obtain the size dependent particle density. For our calculations we estimated an upper and lower uncertainty both for background conditions and during DTDs. For the upper limit we used the ratio between the mean PMcoarse concentration calculated as described in Sec. 2.4 and the mean PMcoarse concentration calculated with the highest density we used of 2.6 g cm−3 .On the other hand, for the lower limit, we used the lowest density of 2.1 g cm−3 for measurements during DTDs and 1.77 g cm−3 for background conditions. During background conditions the aerosol present at Monte Cimone is mainly organics, ammonium sulphate and unknown particles (Putaud et al., 2004) Based on this calculation, we obtained the following uncertainty ranges for the density: DTDs + 9.5 %/ - 9.8 % and background conditions + 11.4 %/ - 28 %. Applying the error propagation, we obtain the upper and lower uncertainty for the PMcoarse concentration during DTDs +/- 61 % and during background conditions + 32 %/ - 41 %.

TREND AND UNCERTAINTY:

We do understand the concern of the reviewer regarding the significance of the trend analysis. We believe that the estimation of the tendency of the fraction of DTDs remains significant, as the changes in detected DTD due to uncertainties are negligible. We are aware that the PMcoarse concentration has some uncertainty, however this does not influence the slope of the trend and we do not expect any changes in the significance.

- UNCERTAINTY IN THE NUMBER OF DETECTED DUST TRANSPORT DAYS
  To assess the uncertainty in the number of the detected dust days, we assumed that the ± 5% uncertainty of the OPC number concentration accounts in the same way for the high frequency component and the threshold values for DTD identification This leads +5 and -8 detected DTDs over a total of 1004 DTDs, which are negligible numbers, that do not have an effect in our presented analysis. We changed the text in L.147 to the following:
  The uncertainty in the quantification of DTDs was calculated assuming the +/- 5% uncertainty of the OPC counting for both the high frequency component and the threshold, which are the variables directly used to identify DTDs. Hence, the maximum overestimation of DTDs was calculated assuming a +5% on the high frequency component and a -5% on the threshold. The opposite was done to

estimate the maximum underestimation of DTDs. Overall, we obtained +5 and -8 DTDs, which are negligible numbers given the total number of 1004 days. We can conclude that an effect of the measurement uncertainty on the analysis presented in the paper can be excluded.

- UNCERTAINTY IN THE PMCOARSE CONCENTRATION

  In the original manuscript we showed in Fig. 3 and Fig. 4 the median and 25[th] and 75[th] percentiles. We agree with the reviewer that plotting the uncertainties provides more information. Therefore, we added the average values together with the error bars, defined as +/- 61% in Fig. 3 a) and 4 a). The average values are consistently higher than the median, which points out that the underlying dataset is not normally distributed and contains extreme values driving the average. This is underlined by the fact that the standard deviation is always larger than the average value. In such cases, it is recommended to rely on the median for the data analysis and interpretation. However, we think that we can add value to the discussion by showing the average, highlighting that a few dust transport events with very high PMcoarse concentrations may influence the statistics of the dataset. Concerning this point, we added a discussion in Section 3.3.1 L.204, which now reads as:

  The average of the PMcoarse concentration during DTDs (Fig 4 (a), dark brown line) is consistently higher than the median. In most of the years the error bars of the average, given as +/- 61% include the median value. In exceptional years, such as 2014, the difference between the median and the average can be as high as a factor of 5, while the standard deviation is always higher than the average. These statistics point out that the PMcoarse concentration during DTDs is driven by one or two events per year transporting very high amounts of dust mass towards Monte Cimone and thus leading to a skewed distribution of the PMcoarse concentration. To reduce the weight of extreme events on the multi-decadal time series, it is recommended to rely on the median values for further analysis. All three variables, i.e., the median of the PMcoarse background, the median and the average PMcoarse during DTDs, showed a wave-like profile with a wavelength of about 12 years.

[Figure]

Figure 3. (a) Annual median PMcoarse concentration during (brown) and outside (grey) DTDs. The dark brown line shows the average values during DTDs including error bars, defined as +/- 61 %. The dashed line shows the trend in the PMcoarse concentration during DTDs. (b) Enhancement in the PMcoarse concentration during DTDs. (c) Enhancement factor (EF) of the PMcoarse concentration during DTDs. In all panels, the solid line shows the median values; the shaded area around is the 25th and 75th percentile.

and 3.3.2 L.223, which now reads as follows:

The average of the PMcoarse concentration during DTDs does not follow the seasonal cycle. While the average PMcoarse concentration aligns with the 75th percentile until October, from October to December it grows up to a factor of 5 higher than the 75th percentile. This increase suggests an increasing influence of intense dust transport events on the PMcoarse concentration and rises an issue on how the PMcoarse concentration during DTDs should be assessed statistically. While the median values help identifying recurring conditions or cycles and drawing a climatology over a long time period, averages may underline months containing strong dust transport events and may be used to isolate specific and intense

anomalies. As the focus of this paper is the analysis of the climatology, the following results will be discussed based only on the median values.

[Figure]

Figure 4. (a) Monthly median PMcoarse concentration during (brown) and outside (grey) of DTDs. The dark brown line shows the average values during DTDs including error bars, defined as +/- 61 %. (b) Enhancement in the PMcoarse concentration during DTDs. (c) Enhancement factor (EF) of the PMcoarse concentration during DTDs. In all panels, the solid line shows the median values; the shaded area around is the 25th and 75th percentile.

- UNCERTAINTY IN THE PMCOARSE ENHANCEMENT AND INFLUENCE ON WORLD HEALTH ORGANIZATION THRESHOLD EXCEEDANCE
  The original manuscript showed the PMcoarse enhancement as function of the duration as a bar graph. Based on the discussion around the uncertainties, we chose to modify Fig. 6 and 7 to show each dust transport event, the overall median and the WHO threshold, as function of the DTE duration. The results are summarized in a table. We accounted for the uncertainty by applying the lower and upper uncertainty

on the data and count the respective days that exceed the WHO threshold. The text is changed to the following:

To assess the intensity of DTEs as function of their duration and peak PMcoarse enhancement, we investigated the change of the maximum daily PMcoarse enhancement of the different DTE durations (Fig. 7). The median of the maximum daily PMcoarse enhancement increased steadily with the duration of DTEs from 2.28 µg m−3 for the 1-day events to 19.47 µg m−3 for DTEs lasting at least 4 days (Table 2) This means that the longer the DTE, the more likely it is to reach higher enhancements in the PMcoarse concentrations. The same pattern was observed when dividing the data into the different seasons (Figure 8 and Table 2). The median values of the maximum PMcoarse enhancement stayed consistently below the WHO threshold value of 45 µg m−3. One reason for the observed behavior could be that for longer DTEs a more extensive dust plume reached CMN, which might be connected to a dust storm over the Saharan desert induced by strong winds. Short DTEs might originate from dust transport of the generally dust loaded air over the Sahara without a prior dust storm. Another important point to mark here is that longer lasting DTEs seem to be more likely to occasionally exceed the threshold value given by the WHO, as the fraction of DTEs above the threshold increases from 5.8% for 1-day events to 24.1% for events that last at least four days. To account for the uncertainty of the PMcoarse enhancement and its effect on the threshold exceedance, the +/- 61% uncertainty, as given in Sec. 2.7, is applied on the data and the respective days exceeding the threshold are counted (see Table 2). While the analysis of the full data set (Fig. 6) is based on enough data for each duration class, some of the seasonal data (Fig.7) must be taken carefully as only very few events are available. Independent of the uncertainty, the numbers presented here give only a lower limit as we consider the PMcoarse and not the PM10 concentration and by that exclude some part of the aerosol mass concentration.

[Figure]

Figure 6. Maximum daily PMcoarse enhancement for the four different lengths of DTE. Each point represents one DTE, the black circle the median value and the dashed line the WHO threshold.

[Figure]

Figure 7. Maximum daily PMcoarse enhancement for the different durations of dust transport events. The data are divided into the four seasons: (a) winter, (b) spring, (c) summer and (d) autumn. Each point represents one DTE, the black circle the median value and the dashed line the WHO threshold.

Table 2. Summary of the results from Fig. 6 and 7 indicating the median of the maximum daily PMcoarse enhancement for the four dust transport event duration categories (1 day, 2 days, 3 days and >= 4 days) the median of the maximum daily PMcoarse enhancement. The percentage of how many DTEs exceeded the WHO threshold value is also provided. To account for the uncertainty, the lower and upper uncertainty was applied on the dataset and the respective number of days above the threshold were counted. The number in brackets give the total number of events

| | | 1 day | 2 days | 3 days | >= 4 days |
|---|---|---|---|---|---|
| **all** | Median | 2.28 | 3.93 | 10.31 | 19.47 |
| | Threshold exceedance | 5.8 % | 9.8 % | 21.0 % | 24.1 % |
| | Min. events - max. events (total events) | 7 - 14 (174) | 1 - 13 (92) | 4 - 16 (62) | 11 - 28 (83) |
| **Winter** | Median | 0.80 | 1.22 | 10.66 | 9.19 |
| | Threshold exceedance | 0.0 % | 5.9 % | 33.3 % | 20.0 % |
| | Min. events - max. events (total events) | 0 - 0 (50) | 0 - 1 (17) | 1 - 2 (6) | 2 - 3 (10) |
| **Spring** | Median | 3.65 | 3.81 | 6.82 | 19.48 |
| | Threshold exceedance | 10.4 % | 16.0 % | 16.7 % | 33.3 % |
| | Min. events - max. events (total events) | 4 - 7 (48) | 0 - 5 (25) | 0 - 3 (18) | 4 - 8 (18) |
| **Summer** | Median | 3.81 | 6.78 | 15.35 | 19.53 |
| | Threshold exceedance | 6.8 % | 3.8 % | 20.8 % | 14.7 % |
| | Min. events - max. events (total events) | 2 - 5 (44) | 1 - 2 (26) | 0 - 8 (24) | 0 - 9 (34) |
| **Autumn** | Median | 2.27 | 7.38 | 9.66 | 22.17 |
| | Threshold exceedance | 6.3 % | 12.5 % | 21.4 % | 33.3 % |
| | Min. events- max. events (total events) | 1 - 2 (32) | 0 - 5 (24) | 3 - 3 (14) | 5 - 8 (21) |

- L.96 the FLEXTRA back trajectory configuration is missing. A section (i.e. meteorological inputs, configuration, model description) should be incorporated into the methods, with details on the configuration of the back-trajectories, given that it represents a major factor in determining the DTD. Why do you decrease the 10 days from Duchi et al. (2016) back trajectory to 7 days? Is there any reason? Please explain.

We followed the suggestion of the reviewer and included a subsection in the methods, which is entitled 'FLEXTRA backward trajectories', where we describe the meteorological input data, the model setup and the output. Furthermore, we explain in more detail the definition of the geographical box used to verify the overpass on the Saharan desert and how it differs from Duchi et al., 2016. Given the residence time of 10-100 h of super-micron particles in the atmosphere (Esmen et al., 1967), we decided to use 7-day backward trajectories. The added section is as follows:

2.2 FLEXTRA backward trajectories

3D-backward trajectories were retrieved from the FLEXTRA model (Stohl et al., 1995), which performs the calculations based on the vertical wind. Meteorological data were provided by ECMWF with a 1.25° x 1.25° grid resolution on 60 vertical levels, derived from a combination of observations with numerical models. In this study, a 7-days long backward trajectory was calculated every 6 h (00, 06, 12, 18 UTC). The initializing height was set to 2200 m a.s.l. and every 3 h the calculation provided several parameters,

among which the location and the altitude of the air parcel. Stohl and Seibert (1998) indicate an accuracy in terms of travel distance around 20 %.

From the location of the air parcels, it can be assessed whether the trajectories traveled over the Saharan desert before reaching Monte Cimone. Therefore, we divided northern Africa into 4 boxes (Fig. 2 c) with the following boundaries:

– Box 1 (Western Sahara): 15 °N to 35 °N and -17 °E to -7 °E

– Box 2 (Central Sahara): 15 °N to 37.5 °N and -7 °E to 15 °E

– Box 3 (Eastern Sahara): 15 °N to 33 °N and 15 °E to 34 °E

– Box 4 (Sahel zone): 10 °N to 15 °N and -17 °E to 34 °E

This grid presents a modified version compared to the one applied in Duchi et al. (2016), where they used one large box ranging from 10 °N to 35 °N and -15 °E to 30 °E. With the new division we fully incorporate the northern part of central Africa and enlarge the included part of the eastern Sahara.

**Minor Comments**

- I would rather suggest adding a section in the introduction on aerosol optical depth as a proxy for dust outbreaks, as it represents a more informative parameter compared to surface PM measurements.

Following the comment we now added a short very general subparagraph in L. 38ff

As summarized by Dulac et al. (2023), there is a long-standing history of Saharan dust characterization and event identification across the entire Mediterranean basin with both in-situ and remote sensing observations. In-situ observations of particulate matter (PM) and aerosol number concentration have been used for more than 30 years to identify the impact of African dust outbreaks on PM levels in the Mediterranean. An increase in the PM concentration has been observed in the upper levels of the atmosphere, but also on the surface at ground level.

And a more detailed description in L.48ff

While in-situ measurements provide direct information on PM concentrations and health-relevant metrics at ground level, remote sensing techniques, both satellite and ground-based, offer broader spatial and temporal coverage on the vertical atmospheric column. The majority of remote sensing-based studies for aerosol-type classification over the Mediterranean were based on sun photometer retrievals like aerosol optical depth (AOD) and its spectral dependence. While satellite remote sensing allows the detection of dust events on regional scale (Barnaba and Gobbi, 2004; Cuevas-Agulló et al., 2024), ground based remote sensing offers continuous observations at local scales, in the eastern (Kaskaoutis et al., 2012;

Kosmopoulos et al., 2008), central (Gobbi et al., 2019; Tafuro et al., 2006) and western (Benkhalifa et al., 2017; Valenzuela et al., 2014) Mediterranean.

- Please consider adding a few sentences describing the structure of the paper at the end of the introduction.

To better point out the structure of the paper, we adjusted the text in L.58ff, which now reads:

This work aims at extending the work from Duchi et al. (2016) until 2023, which allows to investigate not only the annual and interannual variability in dust transport days and particulate matter concentration, but also trends over two decades (see Sec. 3.2 and 3.3). In Sec. 3.3 we further discuss the enhancement in the particulate matter concentration due to transported dust. At the end of the paper we elaborate the duration of dust transport events throughout the months and their intensity based on the enhancement in the particulate matter concentration (see Sec. 3.4 and 3.5).

L.37 Please add a reference

We added the Ginoux et al. (2001) and Sunnu et al. (2008) references, which also occur later in the text.

L.38 if dust particles remain in the upper layers there will be no increase in the surface PM concentration.

We agree that it might be needed to differentiate between the PM concentration on the ground and at higher levels in the atmosphere. Therefore, we added the following sentence in L.38:

... increase the particulate matter (PM) concentration in the upper levels of the atmosphere and when reaching ground level also on the surface.

L.76-77 the GRIMM 1.108 starts at 0.3 µm and not at 0.25 µm with the 780 nm operating wavelength

We corrected the mistake accordingly and added the operating wavelength in the text in L. 75

... , with an operating wavelength of 780 nm.

L.81 Please include the number of bins that you consider for the "coarse" mode.

We added a sentence of the bin numbers in L 81

This corresponds to the OPC bin numbers 6 and higher.

L.92-103 I suggest introducing a list of items instead of the text to identify the different steps of the algorithm.

We followed the suggestion of the reviewer. As a similar comment was made later in the technical comments, we refer for the detailed answer to the later part.

L.102 how many months are available in 21 years of data?

Excluding months with a data coverage of less than 50 % we obtain 221 valid months out of 252. We added the following sentence in L. 102:

..., leading to 221 months out 252.

L.105 Why are back trajectories missing? Due to missing meteorological data?

Yes, back-trajectories can be missing due to missing meteorological data. We added a sentence in L. 106

Back-trajectories might not be available due to missing meteorological data.

L.108-110 "consecutive days", how many consecutive days do you consider? I see it in Figure 1 but it should be written also here.

We agree that this should be mentioned in the text again. Therefore, we added a sentence in L. 110

For the analysis in Section 3.4 and 3.5 the DTEs were split into durations of 1 day, 2 days, 3 days and 4 or more days.

L.117 Please add more details, which is the average particle density you obtain?

We agree that this information was missing and added a respective statement in L. 118

Given that, the particle density ranges from 2.1 g cm-3 to 2.6  g cm-3 for 1 µm and 20 µm particles, respectively, with an average density of 2.4 g cm-3.

L.138 "user-defined alpha value", which is?

We added the alpha value used for the trend analysis in the text in L. 138

... , which in this study is 0.95.

L.168 replace "merging" by "grouping" and "years" by "DTD yearly values".

We changed L.168 as suggested by the reviewer.

L.168-170 here it could be very interesting to compare with the aerosol optical depth. Your results are consistent with the seasonal cycle in the aerosol optical depth climatology observed for the Po Valley in (Di Antonio et al, 2023) using satellite data.

https://acp.copernicus.org/articles/23/12455/2023/

We were not aware of the suggested paper, and included the sentence on how their results compare to ours in the manuscript. Including AOD values of Monte Cimone and a wider area around the station would be subject of a follow up publication focused on reanalysis and remote sensing data. Given the length of our dataset, we prefer to split it in two individual publications rather than having one very long publication showing everything. As also suggested by Reviewer 2, we added 6 years of absorption measurements to compare the optical properties of the dust aerosol in the manuscript. The text in our paper changed to:

A modeling study of the aerosol index by Israelevich et al. (2002), and an analysis of the aerosol optical depth presented by Di Antonio et al. (2023), both suggest an annual cycle with higher values over the Mediterranean region, linked to dust transport.

L.199-200 which is the uncertainty linked to this value?

As suggested by Reviewer 1 and 2, we added a detailed section in the Methods discussing the uncertainties connected to the data analysis presented here. When talking about PMcoarse concentrations, we added the uncertainty throughout the whole text. Furthermore, we included the uncertainty graphically in figures 3 and 4. For the detailed changes in the text regarding the discussion of the uncertainty, we refer to the answer of Specific comment 4.

L.226 Background concentrations are also expected to show diurnal variability in summer compared to winter.

We agree with the reviewer that background concentrations underlay, especially in summer, a diel (total cycle over 24 h) variability. Black carbon is a good proxy for diel variability and in Marinoni et al., 2008, it was shown that the black carbon measured at Monte Cimone follows a diel cycle especially in spring, summer and autumn. In the same study, it is shown that also the fine particle concentration, measured with the OPC 1.108, showed a diel profile. Compared to that, the coarse particle concentration, derived from the same instrument and in the same way as in our study, did not show a diel cycle in any of the seasons. Therefore, the PMcoarse concentrations shown in our paper are representative of the whole day. This means that an increased uncertainty due to a diel cycle is not considered. To make this clear, we added a statement in the text in L.226.

The influence of eventual diel changes in the background concentration due to an increase in the PBL height are negligible, as the coarse particle concentration measured at CMN does not underlay a diel cycle (Marinoni et al., 2008)

L.218-239 What is the key message then here? What do we learn with the seasonal cycle? It is not very clear to me.

We understand that this point was risen in the view of a potential diel cycle in the PMcoarse concentration. As discussed before, we can exclude a diel pattern in the coarse concentration measured at Monte Cimone. Therefore, the discussion of the seasonal cycle provides useful information and shows the reader how the PMcoarse concentration changes throughout the year. By adding the information on the average values of the PMcoarse concentration, we now point out that the median and average can be far from each other, raising the question on how to treat DTDs statistically. While the median provides more information on the overall climatology, the average can be used to identify months with extreme events. Furthermore, this is the first study to show by how much a dust transport event in winter can increase the PM concentration, while in summer this increase is often very small. Even if dust events in winter are rarer, their influence on the PM concentration is much stronger.

L.242-243 this should go in the methods

As suggested, we removed L. 242 to 244 and adjusted the text in L 245. The text was changed to the following:

The interannual variability of the duration of the DTEs, presented as the fraction of the number of DTEs for each duration group over the total number of DTEs in the respective year or month, showed fluctuations throughout the 21 years, but no distinct pattern (Fig. 5a)

L.279 the median is always below the WMO threshold.

We agree with the reviewer that the median values always stay below the threshold. What we intended to say is that during longer dust transport events the threshold is exceeded more often than during shorter events, as the whiskers in Fig. 7 for the 1 and 2 day events never cross the WHO threshold. We adjusted the text in L.274 to

..., but stayed consistently below the WHO threshold value of 45 $\unit{\mu g \ m^{-3}}$

And L. 279 to

... it seems more likely that the PMcoarse enhancement occasionally exceeds the threshold value given by the WHO

L.286-87 I would avoid to make such a suggestion based on the analysis of a single point data source.

We agree with the reviewer that this statement can be misleading and should be based on more data sources. We therefore removed it from the text.

L.297-301 I do not fully agree with this statement. On the one hand, dust transport over the Mediterranean basin is generally favoured during the summer months due to the development of a deeper planetary boundary layer (PBL) over the Sahara, on the other hand, it is also strongly influenced by synoptic-scale weather patterns that facilitate such transport. I would rather say that DTE appears to be more closely associated with the persistence of high-pressure systems over the region, rather than directly linked to PBL development over the Sahara. Given the considerable distance from the source areas, the observations primarily reflect atmospheric transport processes rather than continuous Saharan emissions. It may be helpful to investigate reanalysis data to assess whether the occurrence of longer DTEs is associated with more specific stable atmospheric conditions (looking at geopotential height for example).

We agree that our list of potential causes of longer DTEs is not fully clear, and we did not express strong enough under point (iii) that this is connected to persistent high pressure systems. longer DTEs occur due to two main reasons which are interconnected. First of all, in the summer months, very persistent high pressure systems can form over the Mediterranean, which favor dust transport. Second, the dust mobilization in summer over the Sahara is enhanced and together with the increased PBL height, dust is more easily injected into higher altitudes where it is then transported under higher pressure systems towards Italy. The investigation of reanalysis data could help in answering the question of longer DTEs during summer. As matter of fact, the combination of the in-situ measurements with reanalysis data is in progress and will

be object of a second publication. To make the argumentation in the text clearer, we changed L.297 ff to the following:

Longer lasting events in summer than in winter result from a combination of favorable conditions. In summer, the dust mobilization over the Saharan desert is increased (Vandenbussche et al., 2020; Mousavi et al., 2023) and with the increased PBL height an enhanced dust load is injected to higher altitudes, where it can be more easily transported over long distances (Merdji et al., 2023). In combination with very persistent high-pressure systems, which can form over the Mediterranean in summer, dust transport for a longer time period is favored in the summer months.

L.314 aerosol or dust optical depth?

We meant the dust optical depth and changed the text accordingly.

L.314 what do you mean with "reanalysis data from satellites"?

We agree that this statement is not formulated clear, and we changed the text to the following:

... (iii) investigating the various lengths of DTEs by using the dust optical depth or reanalysis data of the geopotential height.

**Technical comments**

Methods and results can be presented in a clearer way:

- Could you kindly list the different key processes that the dataset has undergone (i.e. Sec. 2.2, 2.3, 2.4) rather than describing them in a block of text?"

To maintain visual coherence in the text formatting, we kept the description of the data treatment procedure as a plain text in former Sec. 2.2. Nonetheless, to improve the understanding of the methodology we provide a table in the supplementary material. In former Sec. 2.3 we added the equation for the calculation of the PMcoarse concentration to improve readability.

The text in L. 91 reads now as:

The Duchi et al. (2016) approach consists of the following steps: (i) 24 h average of the coarse particle number concentration measured with the OPC, (ii) 21 days moving average applied 3 times to dampen the noise. (iii) Subtraction of the third iteration of the moving average from the 24 h average time series to obtain the 'high frequency' (HF) component, (iv) Flag days on which the HF component is above the 95 % confidence interval of all HF components as potential DTDs (v) If any of the trajectory points on the potential DTD passed over the grid specified in 2.2, this day is flagged as a DTD.

The text in L. 115 changed to:

$$\text{PMcoarse} = \sum_i \rho_i \cdot V_i \cdot C_{n,i} = \sum_i \rho_i \cdot \pi/6 \cdot d_i^3 \cdot C_{n,i}$$

Hereby, $C_{n,i}$ is the particle number concentration of the individual bins of the OPC. The volume $V_i$ of the particles with a diameter di is derived from the volume of a sphere, assuming the particle sphericity. The particle density depends on the particle size and composition. Therefore we applied on our data a particle size dependent density $\rho_i$ as presented in Wittmaack (2002). Given that, the particle density ranges from 2.1 g cm−3 to 2.6 g cm−3 for 1 μm and 20 μm particles, respectively, with an average density of 2.4 g cm−3.

The table in the supplementary material reads as:

Table 1. Summary of the key procedures for the data analysis as described in Section 2.3, 2.4 and 2.5

**Identification of dust transport days (Section 2.3)**

| Step | Instrument | Analysis | Product |
|---|---|---|---|
| 1 | OPC | 24 h average coarse concentration | Time series |
| 2 | OPC | 21 days moving average applied 3 times | Noise dampened time series |
| 3 | OPC | Subtraction third iteration of moving average from 24 h average time series | High frequency component (HF) |
| 4 | OPC | Check when HF component it above 95 % confidence interval of all HF components | Potential dust transport days |
| 5 | OPC + back-trajectories | Check if back-trajectories of potential dust days passed over the Saharan desert | Final list of dust transport days |

**Calculation of the PMcoarse concentration (Section 2.4)**

| Step | Instrument | Analysis | Product |
|---|---|---|---|
| 1 | OPC | Calculation of PMcoarse concentration using a particle size dependent density | Time series |

**Calculation of the PMcoarse enhancement (Section 2.5)**

| Step | Instrument | Analysis | Product |
|---|---|---|---|
| 1 | OPC | 30 days moving average of the background PMcoarse concentration | Time series |
| 2 | OPC | Difference between PMcoarse concentration during dust transport days and the background PMcoarse concentration | PMcoarse enhancement |
| 3 | OPC | Fraction between PMcoarse enhancement and background | Enhancement factor (EF) |

- You can summarize major results (such as the average conditions for background/non-background) in tables.

We inserted after section 3.1 a table, which summarizes the major results of Fig.3, Fig.4. and Fig. 5.

Table 1. Summary of major results discussed in Sec. 3.2 and 3.4. Column 1 refers to the different variables, column 2 to the minimum and maximum values of the interannual variability of the respective variables and column 3 to the minimum and maximum values in the annual cycle. For the annual cycle the months in which the minimum and maximum are reached are indicated. Column 4 indicates in which figure the results can be seen.

| | min - max (Interannual variability) | min - max (Annual cycle) | Figure |
|---|---|---|---|
| DTD fraction | 12 % - 20 % | 6 % (Dec) - 19.5 % (Jun) | Fig. 2 |
| PMcoarse background | 0.3 µg m−3 - 3 µg m−3 | 0.2 µg m−3 (Jan, Dec) - 3 µg m−3 (Jul) | Fig. 3a and 4a |
| PMcoarse dust | 2 µg m−3 - 30 µg m−3 | 1 µg m−3 (Jan, Dec) - 10 µg m−3 (May, Jun, Jul) | Fig. 3a and 4a |
| PMcoarse enhancement | 2 µg m−3 - 33 µg m−3 | 1 µg m−3 (Jan, Dec) - 8 µg m−3 (May) | Fig. 3b and 4b |
| EF | 3 - 16 | 2 (Aug) - 25 (Nov) | Fig. 3c and 4c |

L.40 replace "mineral aerosol" by "dust particles"

We changed the text accordingly

L.40 PM "surface" values

We changed the text accordingly

Figures: Adding markers to the line can improve the clarity of the plot.

We followed the suggestion of the reviewer and added in Fig. 4 and 5 markers to the line to improve the clarity.

L.263 add space

We added a space

Literature

Esmen, N. A. and Corn, M.: Residence time of particles in urban air, Atmospheric Environment (1967), 5, 571–578, https://doi.org/https://doi.org/10.1016/0004-6981(71)90113-2, 1971.

Marinoni, A., Cristofanelli, P., Calzolari, F., Roccato, F., Bonafè, U., and Bonasoni, P.: Continuous measurements of aerosol physical parameters at the Mt. Cimone GAW Station (2165 m asl, Italy), 391, 241–251, https://doi.org/10.1016/j.scitotenv.2007.10.004, 2008

---

## Author Comment (AC2)

The paper presents 21 years (2003-2023) of aerosol optical size distribution in a WMO/GAW, ACTRIS-RI and ICOS-RI sampling place at Monte Cimone peak, which have been used to identify dust transport days on site and to analyze its interannual and seasonal patterns for both frequency of occurrence and observed particle concentration. Air masses pathways were also analyzed to confirm the selected days. The paper is an extension of the analysis shows in Duchi et al. (2016), which similar goals and methodology but with 11 years more of database.

We thank Reviewer 2 for the detailed and useful comments. Based on that we made the following main changes in the manuscript:

- Added a section of detailed description of the backward trajectories
- Added a section discussing the uncertainties
- Added a section comparing our work to Duchi et al. (2016)
- Added two tables to summarize the results
- Added supplementary material

The structure of our answer to the comments is: black – Comment of the reviewer, blue – Author's response, red – changes made in the manuscript

**General comments**

The topic of the paper is suitable for ACP, the methodology is clear, the results are logically interpreted and the manuscript is well written. But, it is well-known the doubts of the scientific community when using the aerosol optical size distribution to study desert dust episodes. Therefore, it is observed a lack of explaining the uncertainty in the experimental technique, and how this may affect the estimated parameters. While information of the inventory and duration of the events is strong, other parameters as total mass concentration, enhancement or the comparison with the WHO threshold are subject to some uncertainty. It is due to, as authors comment in Sections 2.1 and 2.3, OPS instruments measure the particle number size distribution, which is converted into a mass distribution using a particle density dependent by particle size, but it also depends on aerosol composition. The dependence on particles composition becomes much stronger, due to the variability in the aerosol typing studied in this work: desert dust and background aerosol.

Therefore, before publishing this work, the authors should resolve this issue, explaining the uncertainties in the estimations. Perhaps, the use of complementary information such as off-line mass concentration, multi-wavelength absorption coefficient, and/or aerosol optical depth could be useful, event if it is of shorter periods.

A similar point was raised by reviewer 1, so we included in the method section a detailed paragraph on the instrumental uncertainties. Moreover, their influence on the data is discussed in multiple parts throughout the manuscript. In the following we list the changes made in the manuscript:

- UNCERTAINTY DEFINITION: we added in the methods a section to describe the uncertainties of the individual terms in the calculation of the PMcoarse concentration. The new section in the methods reads as follows:

2.7 Uncertainties

The calculation of the PMcoarse concentration is subject to uncertainties. Given Equation 1, individual uncertainties of the particle diameter ($d_i$), the particle number concentration ($C_{n,i}$) and the particle density ($\rho_i$), are propagated into a final uncertainty of the PMcoarse concentration.

For $C_{n,i}$, the manufacturer provides for the same OPC model an uncertainty between 3% and 5%. In the literature, the characterization of the uncertainty is limited to one study by Burkart et al. (2010) who observed a 9 % higher total number concentration measured by the same OPC compared to a differential mobility analyzer. However, they did not convert the electrical mobility diameter to an optical equivalent diameter, which can lead to an increased uncertainty. We therefore apply in our calculation of the error propagation the uncertainty of 5 %.

An uncertainty for $d_i$ is not provided by the manufacturer of the OPC; however, it should be accounted due to biases in the correct sizing introduced by non-spheric particles. Putaud et al. (2004) suggest in their study at Monte Cimone a particle sizing uncertainty of 10 % outside of DTDs and of 20 % during DTDs. The higher uncertainty during DTDs arises from the high degree of non-sphericity of dust particles.

The uncertainty for ($\rho_i$) is not given in the study by Wittmaack (2002), which we used to obtain the size dependent particle density. For our calculations we estimated an upper and lower uncertainty both for background conditions and during DTDs. For the upper limit we used the ratio between the mean PMcoarse concentration calculated as described in Sec. 2.4 and the mean PMcoarse concentration calculated with the highest density we used of 2.6 g cm−3. On the other hand, for the lower limit, we used the lowest density of 2.1 g cm−3 for measurements during DTDs and 1.77 g cm−3 for background conditions. During background conditions the aerosol present at Monte Cimone is mainly organics, ammonium sulphate and unknown particles (Putaud et al., 2004) Based on this calculation, we obtained the following uncertainty ranges for the density: DTDs + 9.5 %/ - 9.8 % and background conditions + 11.4 %/ - 28 %. Applying the error propagation, we obtain the upper and lower uncertainty for the PMcoarse concentration during DTDs +/- 61 % and during background conditions + 32 %/ - 41 %.

- UNCERTAINTY IN THE NUMBER OF DETECTED DUST TRANSPORT DAYS
  To assess the uncertainty in the number of the detected dust days, we assumed that the ± 5% uncertainty of the OPC number concentration accounts in the same way for the high frequency component and the threshold value for DTD identification This leads +5 and -8 detected DTDs over a total of 1004 DTDs, which are negligible

numbers, that do not have an effect in our presented analysis. We changed the text in L.147 to the following:

The uncertainty in the quantification of DTDs was calculated assuming the +/- 5% uncertainty of the OPC counting for both the high frequency component and the threshold, which are the variables directly used to identify DTDs. Hence, the maximum overestimation of DTDs was calculated assuming a +5% on the high frequency component and a -5% on the threshold. The opposite was done to estimate the maximum underestimation of DTDs. Overall, we obtained +5 and -8 DTDs, which are negligible numbers given the total number of 1004 days. We can conclude that an effect of the measurement uncertainty on the analysis presented in the paper can be excluded.

- UNCERTAINTY IN THE PMCOARSE CONCENTRATION
  In the original manuscript we showed in Fig. 3 and Fig. 4 the median and 25th and 75th percentiles. We agree with the reviewer that plotting the uncertainties provides more information. Therefore, we added the average values together with the error bars, defined as +/- 61% in Fig. 3 a) and 4 a). The average values are consistently higher than the median, which points out that the underlying dataset is not normally distributed and contains extreme values driving the average. This is underlined by the fact that the standard deviation is always larger than the average value. In such cases, it is recommended to rely on the median for the data analysis and interpretation. However, we think that we can add value to the discussion by showing the average, highlighting that a few dust transport events with very high PMcoarse concentrations may influence the statistics of the dataset. Concerning this point, we added a discussion in Section 3.3.1 L.204, which now reads as:

  The average of the PMcoarse concentration during DTDs (Fig 4 (a), dark brown line) is consistently higher than the median. In most of the years the error bars of the average, given as +/- 61% include the median value. In exceptional years, such as 2014, the difference between the median and the average can be as high as a factor of 5, while the standard deviation is always higher than the average. These statistics point out that the PMcoarse concentration during DTDs is driven by one or two events per year transporting very high amounts of dust mass towards Monte Cimone and thus leading to a skewed distribution of the PMcoarse concentration. To reduce the weight of extreme events on the multi-decadal time series, it is recommended to rely on the median values for further analysis. All three variables, i.e., the median of the PMcoarse background, the median and the average PMcoarse during DTDs, showed a wave-like profile with a wavelength of about 12 years.

[Figure]

Figure 3. (a) Annual median PMcoarse concentration during (brown) and outside (grey) DTDs. The dark brown line shows the average values during DTDs including error bars, defined as +/- 61 %. The dashed line shows the trend in the PMcoarse concentration during DTDs. (b) Enhancement in the PMcoarse concentration during DTDs. (c) Enhancement factor (EF) of the PMcoarse concentration during DTDs. In all panels, the solid line shows the median values; the shaded area around is the 25th and 75th percentile.

and 3.3.2 L.223, which now reads as:

The average of the PMcoarse concentration during DTDs does not follow the seasonal cycle. While the average PMcoarse concentration aligns with the 75th percentile until October, from October to December it grows up to a factor of 5 higher than the 75th percentile. This increase suggests an increasing influence of intense dust transport events on the PMcoarse concentration and rises an issue on how the PMcoarse concentration during DTDs should be assessed statistically. While the median values help identifying recurring conditions or cycles and drawing a climatology over a long time period, averages may underline months containing strong dust transport events and may be used to isolate specific and intense anomalies. As the focus of this paper is the analysis of the climatology, the following results will be discussed based only on the median values.

[Figure]

Figure 4. (a) Monthly median PMcoarse concentration during (brown) and outside (grey) of DTDs. The dark brown line shows the average values during DTDs including error bars, defined as +/- 61 %. (b) Enhancement in the PMcoarse concentration during DTDs. (c) Enhancement factor (EF) of the PMcoarse concentration during DTDs. In all panels, the solid line shows the median values; the shaded area around is the 25th and 75th percentile.

- UNCERTAINTY IN THE PMCOARSE ENHANCEMENT AND INFLUENCE ON WORLD HEALTH ORGANIZATION THRESHOLD EXCEEDANCE

  The original manuscript showed the maximum daily PMcoarse enhancement as function of the duration as a bar graph. Based on the discussion around the uncertainties, we chose to modify Fig. 6 and 7 to show each dust transport event, the overall median and the WHO threshold, as function of the DTE duration. The results are summarized in a table. We accounted for the uncertainty by applying the lower and upper uncertainty on the data and counting the respective days that exceed the WHO threshold. The text is changed to the following:

To assess the intensity of DTEs as function of their duration and peak PMcoarse enhancement, we investigated the change of the maximum daily PMcoarse enhancement of the different DTE durations (Fig. 7). The median of the maximum daily PMcoarse enhancement increased steadily with the duration of DTEs from 2.28 µg m−3 for the 1-day events to 19.47 µg m−3 for DTEs lasting at least 4 days (Table 2) This means that the longer the DTE, the more likely it is to reach higher enhancements in the PMcoarse concentrations. The same pattern was observed when dividing the data into the different seasons (Figure 8 and Table 2). The median values of the maximum PMcoarse enhancement stayed consistently below the WHO threshold value of 45 µg m−3. One reason for the observed behavior could be that for longer DTEs a more extensive dust plume reached CMN, which might be connected to a dust storm over the Saharan desert induced by strong winds. Short DTEs might originate from dust transport of the generally dust loaded air over the Sahara without a prior dust storm. Another important point to mark here is that longer lasting DTEs seem to be more likely to occasionally exceed the threshold value given by the WHO, as the fraction of DTEs above the threshold increases from 5.8% for 1-day events to 24.1% for events that last at least four days. To account for the uncertainty of the PMcoarse enhancement and its effect on the threshold exceedance, the +/- 61% uncertainty, as given in Sec. 2.7, is applied on the data and the respective days exceeding the threshold are counted (see Table 2). While the analysis of the full data set (Fig. 6) is based on enough data for each duration class, some of the seasonal data (Fig. 7) must be taken carefully as only very few events are available. Independent of the uncertainty, the numbers presented here give only a lower limit as we consider the PMcoarse and not the PM10 concentration and by that exclude some part of the aerosol mass concentration.

[Figure]

Figure 6. Maximum daily PMcoarse enhancement for the four different lengths of DTE. Each point represents one DTE, the black circle the median value and the dashed line the WHO threshold.

[Figure]

Figure 7. Maximum daily PMcoarse enhancement for the different durations of dust transport events. The data are divided into the four seasons: (a) winter, (b) spring, (c) summer and (d) autumn. Each point represents one DTE, the black circle the median value and the dashed line the WHO threshold.

Table 2. Summary of the results from Fig. 6 and 7 indicating the median of the maximum daily PMcoarse enhancement for the four dust transport event duration categories (1 day, 2 days, 3 days and >= 4 days) the median of the maximum daily PMcoarse enhancement. The percentage of how many DTEs exceeded the WHO threshold value is also provided. To account for the uncertainty, the lower and upper uncertainty was applied on the dataset and the respective number of days above the threshold were counted. The number in brackets give the total number of events

| | | 1 day | 2 days | 3 days | >= 4 days |
|---|---|---|---|---|---|
| **all** | Median | 2.28 | 3.93 | 10.31 | 19.47 |
| | Threshold exceedance | 5.8 % | 9.8 % | 21.0 % | 24.1 % |
| | Min. events - max. events (total events) | 7 - 14 (174) | 1 - 13 (92) | 4 - 16 (62) | 11 - 28 (83) |
| **Winter** | Median | 0.80 | 1.22 | 10.66 | 9.19 |
| | Threshold exceedance | 0.0 % | 5.9 % | 33.3 % | 20.0 % |
| | Min. events - max. events (total events) | 0 - 0 (50) | 0 - 1 (17) | 1 - 2 (6) | 2 - 3 (10) |
| **Spring** | Median | 3.65 | 3.81 | 6.82 | 19.48 |
| | Threshold exceedance | 10.4 % | 16.0 % | 16.7 % | 33.3 % |
| | Min. events - max. events (total events) | 4 - 7 (48) | 0 - 5 (25) | 0 - 3 (18) | 4 - 8 (18) |
| **Summer** | Median | 3.81 | 6.78 | 15.35 | 19.53 |
| | Threshold exceedance | 6.8 % | 3.8 % | 20.8 % | 14.7 % |
| | Min. events - max. events (total events) | 2 - 5 (44) | 1 - 2 (26) | 0 - 8 (24) | 0 - 9 (34) |
| **Autumn** | Median | 2.27 | 7.38 | 9.66 | 22.17 |
| | Threshold exceedance | 6.3 % | 12.5 % | 21.4 % | 33.3 % |
| | Min. events- max. events (total events) | 1 - 2 (32) | 0 - 5 (24) | 3 - 3 (14) | 5 - 8 (21) |

Further, the reviewer underlined how particle density may change as function of particle composition.

We are aware, that the particle density also depends on the particle composition. In the case of dust transport events the particles in the coarse size range are dominated by dust aerosol which has a density around 2.6 g cm-3. This confirms the choice of the particle size dependent density which has values between 2.1 g cm-3 and 2.6 g cm-3. During background conditions, the aerosol composition is different and is in the super micrometer range a mixture of mainly organics, ammonium sulphate and an unknown component (Putaud et al., 2004). Therefore, Putaud et al. (2004) suggested a density of 1.77 g cm-3. This density is subject to a high uncertainty, as their results are based on 1month of measurements in summer, and thus other seasons are not considered. Furthermore, it is technically challenging to measure the density of super micrometer particles. In the estimation of the uncertainty for the density we used 1.77 g cm-3 to estimate the lower uncertainty for background conditions. The coarse particle concentration during background conditions is typically very low, such that the uncertainty of the OPC measurements dominates over the uncertainty of the particle density. Given all this, we are aware that the PMcoarse concentration during background conditions has some asymmetric bias, but it is challenging to fully account for it. We included a sentence in the new Sec.2.7 of the revised manuscript.

On the other hand, this paper is an extension of Duchi et al (2016), but there is no comparison of the current results with those of the previous article, nor an explanation of the new results.

Furthermore, this citation is very important for the paper, and is not accessible via the indicated DOI. The authors are suggested to make an effort on it as well.

We thank the reviewer for pointing out that the DOI of Duchi et al. (2016) is not working. We contacted the editorial board of the journal without success and thus decided to provide the URL in the bibliography.

We further agree with the reviewer that a part dedicated to the comparison to Duchi et al. (2016) is missing. While both studies give a general overview of the fraction of dust transport days and its seasonal cycle, they differ in the subsequent analysis of these events. Next to the identification of dust events, Duchi et al. (2016) analyze the source origin of the dust and the relation to the coarse particle number concentration. The use of number concentration, however, limits the comparability with other studies. Hence, in our work we converted the size distribution into PMcoarse mass concentration, to better compare our data to previous studies based on in-situ measurements. Considering the comments of both reviewers, the updated version of our manuscript includes a detailed evaluation of uncertainties, which helps quantifying the reliability of the current approach, which was not detailed in Duchi. Overall, the present manuscript provides a more complete phenomenological context and uncertainty evaluation of dust events in Europe, representing an evolution and not only an extension of Duchi. To point this out we added a paragraph in L.192, which reads as:

Comparison to the study of Duchi et al. (2016)

Duchi et al. (2016) analyzed the dust transport at Mt. Cimone between 2002 and 2012. The dataset in this paper extends this analysis until 2023. Both studies observed an overall fraction of DTDs of 15.7 % or 15.8 %, indicating that the annual fraction of DTDs did not change significantly. Also, the seasonal cycle of DTDs was consistent in both studies, with a broad maximum in spring/summer, a second maximum in October/November, and a minimum in winter. When looking at the duration of DTEs, the highest fraction was always the 1 day duration events with 44 % for Duchi et al. (2016) and 42.2 % in this study. For Duchi et al. (2016) the second highest fraction with 28 % were the 2 day events and further they only report that 8 % of the DTEs lasted more than 5 days. In this study, the fraction of the 2 day events was reduced to 22.3 %. The further duration classification differed slightly, as we categorized differently the DTEs based on their duration. After the discussion of the occurrence of DTDs and the seasonal cycle, Duchi et al. (2016) focused their work on the changes in the coarse particle concentration during DTDs and the source origin from the various parts of the Saharan desert. In our work we discuss the interannual variability and the seasonal cycle of the PMcoarse concentration instead of the coarse particle concentration, so that our results can be more comparable to other studies. Furthermore, we give an estimate of the uncertainty related to this analysis.

**Specific comments**

- Please, consider to include information about the backward-trajectories source within Section 2.

We followed the suggestion of the reviewer and included a subsection in the methods, which is entitled 'FLEXTRA backward trajectories', where we describe the meteorological input data, the model setup and the output. Furthermore, we explain in more detail the definition of the geographical box used to verify the overpass on the Saharan desert and how it differs from Duchi et al., 2016. The added section is as follows:

2.2 FLEXTRA backward trajectories
3D-backward trajectories were retrieved from the FLEXTRA model (Stohl et al., 1995), which performs the calculations based on the vertical wind. Meteorological data were provided by ECMWF with a 1.25° x 1.25° grid resolution on 60 vertical levels, derived from a combination of observations with numerical models. In this study, a 7-days long backward trajectory was calculated every 6 h (00, 06, 12, 18 UTC). The initializing height was set to 2200 m a.s.l. and every 3 h the calculation provided several parameters, among which the location and the altitude of the air parcel. Stohl and Seibert (1998) indicate an accuracy in terms of travel distance around 20 %.
From the location of the air parcels, it can be assessed whether the trajectories traveled over the Saharan desert before reaching Monte Cimone. Therefore, we divided northern Africa into 4 boxes (Fig. 2 c) with the following boundaries:
– Box 1 (Western Sahara): 15 °N to 35 °N and -17 °E to -7 °E
– Box 2 (Central Sahara): 15 °N to 37.5 °N and -7 °E to 15 °E
– Box 3 (Eastern Sahara): 15 °N to 33 °N and 15 °E to 34 °E
– Box 4 (Sahel zone): 10 °N to 15 °N and -17 °E to 34 °E
This grid presents a modified version compared to the one applied in Duchi et al. (2016), where they used one large box ranging from 10 °N to 35 °N and -15 °E to 30 °E. With the new division we fully incorporate the northern part of central Africa and enlarge the included part of the eastern Sahara.

- Figure 1 – the legend about (a), (b) and (c) is not corresponding to the text in paper. Please, revise.

Thank you for noticing this mistake. We corrected the figure caption accordingly.

Figure 2. (a) Fraction of dust transport days (brown) and the number of non-dust transport days (grey). (b) Duration of dust transport events divided into 1 day (beige), 2 days (orange), 3 days (light brown) and 4 and more days (dark brown). (c) Grid box extension for the four boxes used to confirm dust transport days. The percentage values give the f

raction of back-trajectories that passed over each box. Map made with Natural Earth (naturalearthdata.com).

- Lines 162-163: it is commented that there was no significant temporal trend (slope of 0.063) in the fraction of DTDs obtained from the trend analysis. But dashed line shows an increasing trend. Please, consider to delete the line, because this may lead to misinterpretation.

  We agree with the reviewer that the trend shown in the figure can be misleading for the reader. We therefore removed the trend in Fig. 2a as suggested.

- Line 235: opposing - > opposite

  We corrected opposing to opposite as suggested.

- Line 247: in the majority of the years was -> in most years was

  We corrected the text as suggested.

- Line 249: longer of shorter -> longer or shorter

  We corrected the typo

- Lines 262-263: dust mobilization in summer -> dust injection into the atmosphere during the summer.

  We corrected the text as suggested.

- Lines 269-271: care must be taken because the results may have a high uncertainty. It is important to solve this issue.

  We agree with the reviewer that the PMcoarse enhancement is subject of uncertainty. As already pointed out in the general comments we added more information on that throughout the manuscript. We also specifically addressed this part and refer to the made changes to our response to the general comments.

Literature

Putaud, J.-P., Van Dingenen, R., Dell'Acqua, A., Raes, F., Matta, E., Decesari, S., Facchini, M. C., and Fuzzi, S.: Size-segregated aerosol mass closure and chemical composition in Monte Cimone (I) during MINATROC, Atmospheric Chemistry and Physics, 4, 889–902, https://doi.org/10.5194/acp-4-889-2004, 2004.

---

## Author Response (AR2)

I thank the Authors for undertaking a huge work on the manuscript, following Reviewers' recommendations. The paper is now significantly improved and suitable for publication. However, after my own revision, I believe that an important issue still needs to be addressed.

We thank the editor for his comments and provide in the following our detailed response and the respective changes.

The structure of our answer to the comments is: black – Comment of the editor, blue – Author's response, red – changes made in the manuscript

Specifically, I agree with Reviewer #1 regarding the need for a comparison with reanalysis and/or satellite products to provide an assessment of the representativeness of your results. PM concentrations and AOD data are available in the CAMS archive, I recommend adding a comparison with grid points near Mt. Cimone for, at least, the seasonal cycle and the interannual variability.

We followed the recommendation of the editor and downloaded CAMS data from 2003 to 2023. Hereby, we chose the dust mixing ratio of the size range 0.9  $\mu$ m to 20  $\mu$ m, which we then converted with the air density into a mass concentration. We now compare the interannual variability and the seasonal cycle of the PMcoarse concentration from the measurements with the dust mass concentration from the CAMS reanalysis data. Therefore, we added in the methods in L. 164 a section on the CAMS reanalysis data, which reads as follows:

**CAMS** (Copernicus Atmosphere Monitoring Service, https://ads.atmosphere.copernicus.eu/datasets, last accessed 08-10-2025) provides global reanalysis of various atmospheric constituents. The EAC4 (ECMWF Atmospheric Composition Reanalysis 4) reanalysis data are provided for a 0.7° x 0.7° grid with a vertical resolution of 60 hybrid sigma-pressure (model) levels. The time resolution is 3 h. To compare the here presented PMcoarse concentration to reanalysis data, we used the dust aerosol mixing ratio (0.9 - 20 µm), which was converted to a mass concentration using the provided air density for the selected chosen model level. Data were downloaded for all the years (2003 to 2023) for the model level 46, which corresponds to a geometric altitude of 2327.89 m and a pressure of 780.3455 hPa. As CMN is situated between the provided grid points, the mass concentration was averaged over the four closest points. Further, the data were averaged over 24 h to obtain the same time resolution as the measurements.

**In the results in Section 3.4.1 in L. 291 we added:**

Institutions like Copernicus provide dust forecasts based on the aerosol optical depth, which is a variable integrated over the full atmospheric column (Blake et al., 2025). To verify the representativity of CAMS reanalysis with in-situ observations we provide for the first time a comparison of the PM mass concentration on a single level on a long term and local scale. The median mass concentration retrieved from CAMS reanalysis reflects, overall, the interannual

variability of the measurements and falls within the range between the 25th and 75th percentile. Exceptions are years with a very high (2014) or very low (2019) measured PMcoarse concentrations. Potential studies should address especially the underestimation of CAMS during episodes of extreme dust transport events during which high PM concentrations were measured. Further differences between the CAMS reanalysis data and the measurements can originate from the choice of the vertical model level and the horizontal grid and require a dedicated sensitivity study.

**And in Section 3.4.2 in L. 320 we added:**

The CAMS reanalysis reflects the seasonal cycle of the measurements and shows a minimum in the winter months and a maximum in April/May. In the summer and autumn months the CAMS data points fall well within the 25th/75th percentile and are very close to the measured points. This difference is increased especially in winter, where CAMS consistently underestimates the PM mass concentration.

**Figure 3**. (a) Annual median PMcoarse concentration during (brown) and outside (grey) DTDs. The dark brown line shows the average

values during DTDs including error bars, defined as +/- 61 %. The crosses show the median value of the CAMS reanalysis data. The dashed line shows the trend in the PMcoarse concentration during DTDs. (b) Enhancement in the PMcoarse concentration during DTDs. (c) Enhancement factor (EF) of the PMcoarse concentration during DTDs. In all panels, the solid line shows the median values; the shaded area around is the 25th and 75th percentile.

**Figure 4.** (a) Monthly median PMcoarse concentration during (brown) and outside (grey) of DTDs. The dark brown line shows the average values during DTDs including error bars, defined as +/- 61 %. The crosses show the median value of the CAMS reanalysis data. (b) Enhancement in the PMcoarse concentration during DTDs. (c) Enhancement factor (EF) of the PMcoarse concentration during DTDs. In all panels, the solid line shows the median values; the shaded area around is the 25th and 75th percentile.

A few minor issues are listed below.

L32: I'm not aware of any atmospheric structure named 'the Mediterranean cyclone'. You maybe refer to moving Mediterranean cyclones modulating dust transport from North Africa. Please revise this expression here and across the text.

We rephrased all the according lines (L.32, 227, 246, 382, 383) to 'cyclone in the Mediterranean'

L42: what is the difference between surface and ground level?

We agree that the sentence in L. 42 might be misleading. It now reads as follows:

, but also at ground level.

L120-125: please revise punctuation and capitals.

We checked and corrected the punctuation and capitals.

Section 2.2: please add here the motivation for reducing back trajectories to 7 days.

Given the residence time of 10-100 h of super-micron particles in the atmosphere (Esmen et al., 1967), we decided to use 7-day backward trajectories. We added the following sentence in L. 105:

The trajectories were limited to 7 days due to the atmospheric residence time of super-micron particles between 10 h and 100 h (Esmen and Corn, 1971)

L202-206: I'd move those lines to Section 2.7.

We agree with the editor and moved the description of the uncertainty to Section 2.7

L263-267: I suggest moving those considerations to the Conclusions section, to highlight the novelty of the paper.

We would like to keep the statement in these lines, in order to provide a complete comparison to the study by Duchi et al., 2016. As suggested, to highlight the novelty of the paper, we added a similar statement in the conclusions in L. 380:

Additionally, to what was presented in Duchi et al. (2016), we give a detailed evaluation of the uncertainty and present the PMcoarse concentration instead of the number concentration of coarse particles, to be more comparable to other studies.

L382-384: There is no evidence presented in your analysis regarding the relationship between atmospheric circulation and dust transport in the Mediterranean, please make clear that those considerations come from the literature.

We added the according references (Ginoux et al. (2001), Sunnu et al. (2008), Varga (2020) and Flaounas et al. (2022)) in the text.

L393-396: Same as above, please provide some references.

We added the according references (Cristofanelli et al. (2018) and Mifka et al. (2022)) in the text.

In your response there are some mismatches in the figure referencing with respect the revised version, please verify that all the figures are correctly referred in the text.

We verified that all the figures are referred correctly in the text.

With the next revision, please re-name supplement materials according to ACP standards: https://www.atmospheric-chemistry-and-physics.net/submission.html#assets > Supplements

We changed the supplements according to ACP standards

**References:**

Blake, L., Arola, A., Benedictow, A., Bennouna, Y., Bouarar, I., Cuevas, E., Errera, Q., Eskes, H., Griesfeller, J., Ilic, L., Kapsomenakis, J., Langerock, B., Li, C.W. Y., E, Mortier, A., Pison, I., Pitkanen, M., Richter, A., Schoenhardt, A., Schulz, M., Tarniewicz, J., Tsikerdekis, A., Warneke, T., and Zerefos, C.: Validation report for the CAMS global reanalyses of aerosol and reactive trace gases:2003-2024, Copernicus Atmosphere Monitoring Service (CAMS) report, https://doi.org/10.24380/vv0t-8tcg, 2025.